# Monsoon Mission Coupled Forecast System version 2.0: model description and Indian monsoon simulations

**Deepeshkumar Jain**[1,a]**, Suryachandra A. Rao**[1]**, A. Ramu Dandi**TS1[1]**, Prasanth A. Pillai**[1]**, Ankur Srivastava**TS2[1]**, Maheswar Pradhan**[1]**, and V. G. Kiran Gangadharan**TS3[1]

[1]Monsoon Mission, Indian Institute of Tropical Meteorology, Ministry of Earth Sciences, Pashan, Pune, 411008, Maharashtra, India
[a]TS4currently at: NCMRWF, Ministry of Earth Sciences, A50, Noida, 201309, UP, India

**Correspondence:** Suryachandra A. Rao (surya@tropmet.res.in)

**Abstract.** We present the Monsoon Mission Coupled Forecast System version 2.0 (MMCFSv2) model, which substantially upgrades the present operational MMCFSv1 (version 1) at the India Meteorology Department. The latest 25 years (1998–2022) of retrospective seasonal coupled hindcast simulations of the Indian summer monsoon with April initial conditions from Coupled Forecast System Reanalysis are discussed. MMCFSv2 simulates the tropical wind, rainfall, and temperature structure reasonably well. MMCFSv2 captures surface winds well and reduces precipitation biases over land, except over India and North America. The dry bias over these regions remained like in MMCFSv1. MMCFSv2 captures significant features of the Indian monsoon, including the intensity and location of the maximum precipitation centers and the large-scale monsoon circulation. MMCFSv2 improves the phase skill (anomaly correlation coefficient) of the interannual variation of Indian summer monsoon rainfall (ISMR) by 17 % and enhances the amplitude skill (normalized root mean square error) by 20 %. MMCFSv2 shows improved teleconnections of ISMR with the equatorial Indian and Pacific oceans. This 25-year hindcast dataset will serve as the baseline for future sensitivity studies of MMCFSv2.

## 1 Introduction

Over a third of the world's population resides in the east Asian and the Indian sub-continent region, most of which depends on the natural irrigation from the summer monsoon rainfall for agricultural production (Gadgil, 2006). Indian summer monsoon (ISM), which lasts from June to September every year, is a perennial system. It, however, shows interannual and intraseasonal variability (Parthasarthy et al., 1993; Kumar et al., 1999a; Munot et al., 2000; Mohan and Goswami, 2000; Gadgil, 2003) affecting the region's agricultural production (Gadgil, 2006). A 10 % deviation from the climatological mean is sufficient to have an excess or a deficient monsoon over India (Singh et al., 2015). The standard deviation, variance, and their ratio with mean of June to September (JJAS) mean precipitation (Fig. 1) shows that the location of highest variability is over oceans. In contrast, the variability over the Indian landmass is low despite the high mean precipitation. This low variability (having a high impact on agricultural production) challenges the models trying to predict it.

The monsoon is an inherently coupled system (Webster et al., 2002; Ramu et al., 2016), and the Indian Meteorological Department (IMD) has been using the Monsoon Mission Coupled Forecast System version 1 (MMCFSv1) model operationally to predict the ISM since 2011 (Benke et al., 2019). The Indian Institute of Tropical Meteorology (IITM) has been using MMCFSv1 as a research testbed to study the various facets of ISM rainfall (ISMR) (Ramu et al., 2016; Krishna et al., 2019; Srivastava et al., 2021; Pillai et al., 2021; Pradhan et al., 2022; Rao et al., 2019). MMCFSv1 is based on a high-resolution climate forecast system model from the U.S. National Centers for Environmental Prediction (NCEP) (Saha et al., 2014b).

While the NCEP runs the model at a resolution of T126, IITM runs it at T382. Ramu et al. (2016) analyzed both

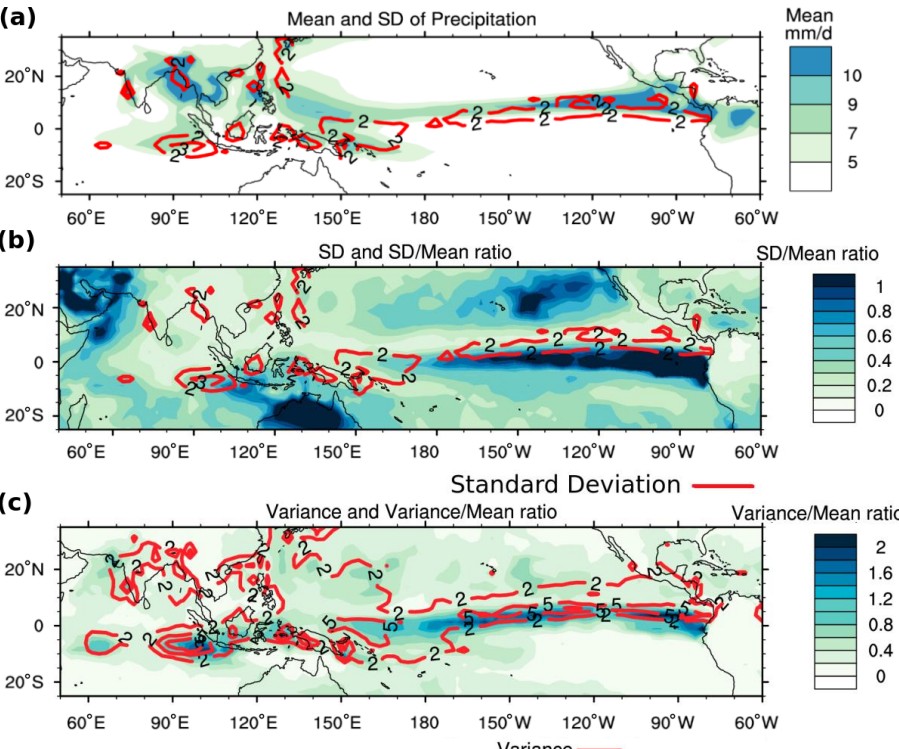

**Figure 1.** Panel **(a)** shows JJAS climatological mean precipitation (shading) and standard deviation (contours), **(b)** shows standard deviation (red contour) and standard deviation to mean ratio (shading), and **(c)** shows variance (red contour) and variance to mean ratio of precipitation (shading) from 41 years (1981–2021) of Global Precipitation Climatology Project data.

the model resolutions (T126 and T382) based on 28 years of hindcast data to show that the skill of lower-resolution model is 0.49 compared with the skill of 0.55 of the higher-resolution model with the February initial conditions. Pillai
et al. (2018b) TS5 have shown that the potential predictability of ISMR is 0.7 in MMCFSv1, and the maximum actual skill (with different initial conditions) of the operational model at IMD (MMCFSv1) is 0.55. Hence, the gap between the potential predictability and actual skill is large. Many factors, such as resolution, initial conditions, physics, and dynamics, limit the models' skill. Coupled climate models' skill improvement involves efforts from many research groups specializing in a particular sub-domain (component) of the coupled model. As its components, the MMCFSv1 has a global forecast system atmospheric model with a Eulerian dynamical core (GFS-EL; Moorthi et al., 2001), Modular Ocean Model version 4 (MOM4; Griffies et al., 2004), and a sea-ice model (SIS; Semtner, 1976; Winton, 2000), all coupled together using a hard-coded coupler. This hard-coded coupler runs on a single core. This presents a computational problem as the models grow in complexity and become highly parallelizable.

With an ever-increasing understanding of our climate system, the above-mentioned individual components of MM-CFSv1 have seen a lot of improvements independently of

each other. MOM6 (Adcroft, 2016) is a significant upgrade (algorithmically) over MOM4 (discussed in detail in the next section). The predictability of the medium-range atmospheric models has improved with increasing model resolutions. The need for higher atmospheric model resolution emphasizes using a semi-Lagrangian dynamical core in place of the Eulerian one (Staniforth et al., 1991). CICE5 (Bailey et al., 2018) is a separate code base designed to be used in coupled models and is highly parallelizable and brings in many improvements (see next section) over the SIS (Semtner, 1976; Winton, 2000) sea-ice model, which is a part of the MOM4 code base.

As mentioned above, the components in MMCFSv1 are hard coded to transfer and transform the data from one process (model component) to another through a coupler. To make any changes to the individual model component, one must understand how these model components are implemented and how the coupler accumulates, transfers, and regrids the boundary condition data from one component to another. However, since the coupler is hard coded to interface with the individual model components, there is a lack of modularity in how MMCFSv1 is implemented.

Realizing that this hinders seamless model development in coupled models, many research groups across the climate community (e.g., Black et al., 2009; Craig et al., 2017; Bal-

aji, 2004) have been developing the software infrastructure in trying to bring modularity to the complicated climate model codes. The National Oceanic and Atmospheric Administration (NOAA) Environmental Modeling System (NEMS) is one such modeling framework (Black et al., 2009) which is used to streamline components of the models. The NEMS architecture is based on the Earth Modeling System Framework (ESMF; Hill et al., 2004). The ESMF standardizes how the model components interact with each other, thus bringing in modularity. The NEMS refines the definition of what it means to be a model component and standardizes the initiation, running, and finalizing steps of each model component. MMCFSv2 uses the NEMS coupling framework and upgrades all the major individual model components of MMCFSv1 (Table 1). Using a NEMS coupler will facilitate easier future upgrading of MMCFSv2 components.

Systematic biases in MMCFSv1 have been well documented (Ramu et al., 2016; Pillai et al., 2017; Chaudhari et al., 2013 TS8), and the significant biases are the cooler sea surface temperature (SST) (especially over the Indian and southern Pacific oceans), dry bias over land, wet bias over the ocean, and weaker monsoon circulations. Hence, in this study, we have investigated the model's ability to simulate the mean state and assessed the model's skill in predicting the phase and amplitude of ISMR. We have limited our simulations to 25 years of retrospective hindcasts (1998–2022) due to limitations in computational resources. The present paper gives details of MMCFSv2 individual component upgrades. We also analyze the simulated mean tropical SST, circulation, and the mean and interannual variability of ISMR as well as its teleconnection with different oceanic modes. Section 2 discusses model upgrades over MMCFSv1. Section 3 describes the experimental design for this study. We then show the simulated results and compare them with MMCFSv1 in Sect. 4 before summarizing them in the last section.

## 2 MMCFSv2 model details

We use the NCEP MMCFSv1 (Saha et al., 2014b) as the base model to discuss the upgrades MMCFSv2 brings. The primary individual model components of MMCFSv1, upgraded in MMCFSv2, are tabulated in Table 1 and discussed briefly below. The MMCFSv1 uses the spectral model Global Forecast System (GFS) as the atmospheric model (Moorthi et al., 2001) with the Eulerian dynamical core. MMCFSv2 instead uses a Semi-Lagrangian dynamical core for the GFS (GFS-SL, Sela, 2010; Mukhopadhyay et al., 2019). Using a Semi-Lagrangian dynamical core allows us to have higher atmospheric model resolutions while keeping the time stepping the same.

### 2.1 MOM6 ocean model

MMCFSv1 uses the Geophysical Fluid Dynamics Laboratory Modular Ocean Model version 4p0d (MOM4) as the ocean model (Griffies et al., 2004). It has been upgraded to MOM6 (Adcroft, 2016) in MMCFSv2. MOM6 is based on generalized ocean dynamics which enables variable vertical and hybrid coordinates. Since MOM6 uses vertical Lagrangian remapping (Griffies et al., 2020), it can be configured with any vertical coordinates among geopotential, isopycnal, terrain following, or hybrid (user defined). Significant improvements brought by MOM6 over MOM4 include using C-grid stencil over B-grid stencil. C-grid stencil is preferred for simulations involving an active mesoscale eddy field. MOM6 uses scale-aware parameterizations for mesoscale eddy-permitting regimes based on Jansen et al. (2019) and eddy fluxes are parameterized based on Jansen et al. (2015). The boundary layer scheme in MOM6 is based on Reichl et al. (2018) and incorporates Langmuir mixing. It also introduces a suite of parameterized mixing from breaking gravity waves. A new method for performing neutral diffusion is also introduced in MOM6 that prevents the spurious formation of extrema. (Complete details of MOM6 can be found at https://www.gfdl.noaa.gov/mom-ocean-model/; last access: 29 December 2023).

The present configuration of MOM6 used in MMCFSv2 consists of 40 vertical levels with HyCOM-like hybrid coordinates. The horizontal grid follows a tripolar grid with poles over land and $1440\times1080$ pixels in the $x$ and $y$ directions, respectively. The resolution of the ocean component in MOM4 is 0.25° between 10° S and 10° N latitude band and 0.5° elsewhere. This has been increased to 0.125° TS9 the Equator in MOM6. MOM6 is compiled with external coupler and a cap code (mom_cap.F90) interfaces the model with the NEMS framework. The coupling happens every 30 min.

### 2.2 CICE5.0 model

MMCFSv1 uses a three-layer (one layer of snow and two layers of sea ice) interactive sea-ice model (Winton, 2000), which is an improvement over the Semtner three-layer model (Semtner, 1976). This component model has been upgraded to the Los Alamos CICE5 (Hunke et al., 2015) in MMCFSv2. CICE5 is designed to be used in coupled models and is highly parallelizable. The major improvements of CICE5 over the sea-ice model of MMCFSv1 include ice velocity in atmosphere–ice coupling updates and allowing a variable coefficient for the ice–ocean heat flux.

CICE5 is developed by LANL to be used in fully coupled climate models. CICE5.0 has (improved or new) parameterizations for form drag, sea-ice biogeochemistry, and explicit melt pond, among others. CICE5 has been extensively used in climate simulations by the Community Earth System Model (CESM). More details can be found in the CICE documentation (link in "Data availability" section). CICE5 runs

**Table 1.** Major changes to model components between MMCFSv1 and MMCFSv2.

| Model/component | Atmosphere (resolution) | Ocean (resolution) | Ice model | Land model | References |
|---|---|---|---|---|---|
| MMCFSv1 | GFS-EL (T382, ~38 km) | MOM4p0d (0.25 TS6 a ... S–10° N) | SIS sea ice | NOAH-LSM | Moorthi et al. (2001) Griffies et al. (2004) Winton (2000) Ek et al. (2003) |
| MMCFSv2 | GFS-SL (T574, ~38 km) | MOM6 (0.125 TS7 a ... S–10° N) | CICE5 | NOAH-LSM | Sela (2010) Adcroft (2016) Hunke et al. (2015) Ek et al. (2003) |
| Parameterizations | Cumulus | Ocean vertical grids | Ocean physical closures | | |
| V1 | SAS | Fixed (B stencil) | Non-scale aware | | |
| V2 | New SAS | Arbitrary Lagrangian Eulerian (C stencil) | Scale-aware parameterizations | | |
| Horizontal grid size | | | | | |
| V1 | 1152 × 576 | 720 × 410 | 720 × 410 | 1152 × 576 | |
| V2 | 1 152 × 576 | 1440 × 1080 | 1440 × 1080 | 1152 × 576 | |

at MOM6 resolution of 1440 × 1080 pixels in the horizontal grid. Similar to MOM6, CICE5 code is coupled to the NEMS framework using a cap code (cice_cap.F90), and the coupling happens every 30 min.

## 2.3 Coupler

The hard coded coupler in MMCFSv1 runs on a single processor and transfers surface fluxes (wind stress and radiative) to the ocean model and provides SST to the atmospheric model after every coupling time step of 1800 s. This coupler has been replaced by the NEMS coupler, which is a parallel coupler and is currently running on 144 cores in MMCFSv2. (More details on the NEMS coupler can be found in Black et al., 2009.)

## 2.4 Other model components

As mentioned earlier, the atmospheric component is based on GFS with semi-Lagrangian dynamical core. Although the four-layer NOAH land surface model (Ek et al., 2003) remains the same between MMCFSv1 and MMCFSv2, the NEMS framework allows us to include newer versions of land models such as NOAH-MP. This will be done in future work.

## 3 Experimental details and observational/reanalysis data

The retrospective ensemble prediction (hindcast) runs of the MMCFSv1 have atmospheric horizontal resolutions corre-

sponding to triangular truncation of T382L64, while that of MMCFSv2 is T574L64 (horizontal resolution of both versions is ~38 km). The atmosphere, land, and ocean initial conditions for these runs are obtained from the NCEP Climate Forecast System Reanalysis (CFSR) (Saha et al., 2010). The atmospheric component of MMCFSv1 and MMCFSv2 has 64 sigma-pressure hybrid vertical levels and the ocean component has 40 vertical layers. The convective parameterization scheme used in the atmospheric part of MMCFSv1 and MMCFSv2 is based on the Arakawa–Schubert scheme, with orographic gravity wave, drag, and momentum mixing.

Pillai et al. (2022) showed that the prediction skill for El Niño–Southern Oscillation (ENSO) was lower for MMCFSv1 initialized with February (3-month lead time) initial conditions compared with when it was initialized with April (1-month lead time) initial conditions. They showed that models which depend on ENSO teleconnection for ISMR interannual variability (MMCFSv1 in their case) have better ISMR prediction skills with April initial conditions. Hence, the MMCFSv2 experimental setup is based on a 10-member lagged ensemble with April initial conditions (00z01Apr, 12z01Apr, 00z06Apr, 12z06Apr, 00z11Apr, 12z11Apr, 00z16Apr, 12z16Apr, 00z21Apr, and 12z21Apr), while that of MMCFSv1 is like the one in Ramu et al. (2016), albeit for April initial conditions and a total of 12 ensembles (2 additional ensembles corresponding to 00:00 and 12:00 Z of 26 April). Each hindcast run is integrated for 6 months, from April to September. A total of 25 years (1998–2022) of hindcasts have been performed.

For the verification of the model-simulated rainfall, we use the Global Precipitation Climatology Project (GPCP; Adler et al., 2003) and 1° gridded daily rainfall from IMD (Rajeevan et al., 2006) for the same hindcast period (1998–2022). It is noteworthy that IMD uses data from variable rain gauge networks from day to day based on the availability of data from gauges. However, GPCP uses data from a fixed rain gauge network. Since IMD keeps updating the rain gauge network continuously, the seasonal mean values also vary for each update (Pai et al., 2014). Hence, in this study, we use GPCP data as a standard product for assessing the skills of the ISMR. For SST validation, we use the Extended Reconstructed Sea Surface Temperature (ERSST) version 5 (Huang et al., 2017, 2018, 2019, 2020). In addition, we use ERA5 reanalysis products for winds (Hersbach et al., 2020). Model simulated mixed layer depth (MLD) was compared with the CMCC Global Ocean Physical Reanalysis System (C-GLORS) MLD. We have used Pearson's test to compute the statistical significance of correlation coefficients and Student's $t$ test to compute the statistical significance of difference. Anomaly correlation coefficients (phase skill) and normalized root mean square error (normalized with standard deviation; amplitude skill) are used as a metric to assess the skill of the models in capturing ISMR and other tropical interannual modes.

## 4 Results

We first look at the ability of MMCFSv2 to simulate the mean tropical climatology of rainfall and surface temperature. We then look at the simulated large-scale circulation and interannual variability of ISMR before examining its teleconnections with different interannual modes in the tropics.

### 4.1 Climatology

#### 4.1.1 Mean rainfall

Most climate models (Pillai et al., 2018a; Sabeerali et al., 2013) have shown that land rainfall is underestimated, while rainfall over oceans is overestimated. Figure 2 shows JJAS mean precipitation from GPCP (observed) and the models. GPCP (Fig. 2a) shows maximum rainfall over a band along the tropical Pacific Ocean. Both models simulate this tropical rain belt (Fig. 2b, c) reasonably well. Surprisingly, the dry bias over land, which is normally present in many of the climate models, is absent in MMCFSv2, except over the Indian land region. While the rainfall dry bias over west central India is increased, the bias over northeast and east central India is reduced. MMCFSv1 has significant wet bias over the North Pacific, Atlantic, and eastern Indian oceans regions (Fig. 2d) which is significantly reduced in MMCFSv2 (Fig. 2e, f). MMCFSv2 is also closer to GPCP in Africa and South American regions.

Over the ISM region, there are three locations of precipitation maximum, viz. the head Bay of Bengal, the Western Ghats, and the southeastern equatorial Indian Ocean (Fig. 2a). Both models get these precipitation maxima (Fig. 2b, c). There is a strong wet bias over the Indian Ocean basins and a dry bias over the northwest Indian landmass in MMCFSv1 (Fig. 2d). This is significantly reduced in MMCFSv2. The dry bias over the Indian landmass seen in both models is consistent with previous studies (Goswami et al., 2014; Saha et al., 2014a; George et al., 2016; Ramu et al., 2016; Pillai et al., 2018a). A study by Sabeerali et al. (2013) has reported similar precipitation bias in many CMIP5 models. Nevertheless, MMCFSv2 improves the dry bias over the Indian landmass over MMCFSv1 (Table 2). From recent CMIP6 models, the majority of the models also suffer from similar rainfall biases to those of MMCFSv1 (Choudhury et al., 2021 TS10).

#### 4.1.2 Temperature bias

The spatial distribution of observed (ERSST) and simulated (MMCFS) climatological JJAS mean SST is shown in Fig. 3. The presence of equatorial maxima characterizes the observed SST. The SST over the Indo-Pacific region is greater than 28 °C and is known as the Indo-Pacific warm pool region (ERSST in Fig. 3a). Both models can simulate the large-scale distribution of tropical SST (Fig. 3b, c). MMCFSv1 shows a cold bias (greater than 0.5 °C) over the tropical Indian Ocean and southern Pacific (Fig. 3d). This cold bias has been reported previously by many studies (George et al., 2016; Pokhrel et al., 2012; Saha et al., 2014a). It has been shown by Pokhrel et al. (2012) to be due to the dry surface atmosphere and an associated increase in latent heat flux in MMCFSv1. MMCFSv1 also has a strong warm bias (greater than 0.5–1.5 °C) over the northwestern, southwestern, northeastern, and southeastern Pacific (Fig. 3d). Zheng et al. (2011) reported that this strong warm bias over the northeastern and southeastern Pacific is due to the misrepresentation of stratus cloud decks and an associated increase in incoming shortwave radiation flux.

The cold SST bias of MMCFSv1 over the Indian Ocean is significantly reduced in MMCFSv2 (Fig. 3e). MMCFSv2 has a warm bias (greater than 0.5 °C) over the entire Indian Ocean except for the extreme southeast IO and the northern Arabian Sea (Fig. 3e). The warm biases are intensified over the Pacific region, except the southeastern Pacific in MMCFSv2 compared with MMCFSv1 (Fig. 3f). Overall, there is a warming of SST over the tropics in MMCFSv2 compared with MMCFSv1. In fact, the latest CMIP6 models also have similar warm biases in SST (Farneti et al., 2022).

MMCFSv1 underestimates surface air (2 m) temperature (Fig. 7) over most of the land, including the Tibetan Plateau, except the African region (overestimation by 2–4 °C). MMCFSv2 overestimates surface air temperature over most of the tropics by more than 3 °C. The warm SST bias in the trop-

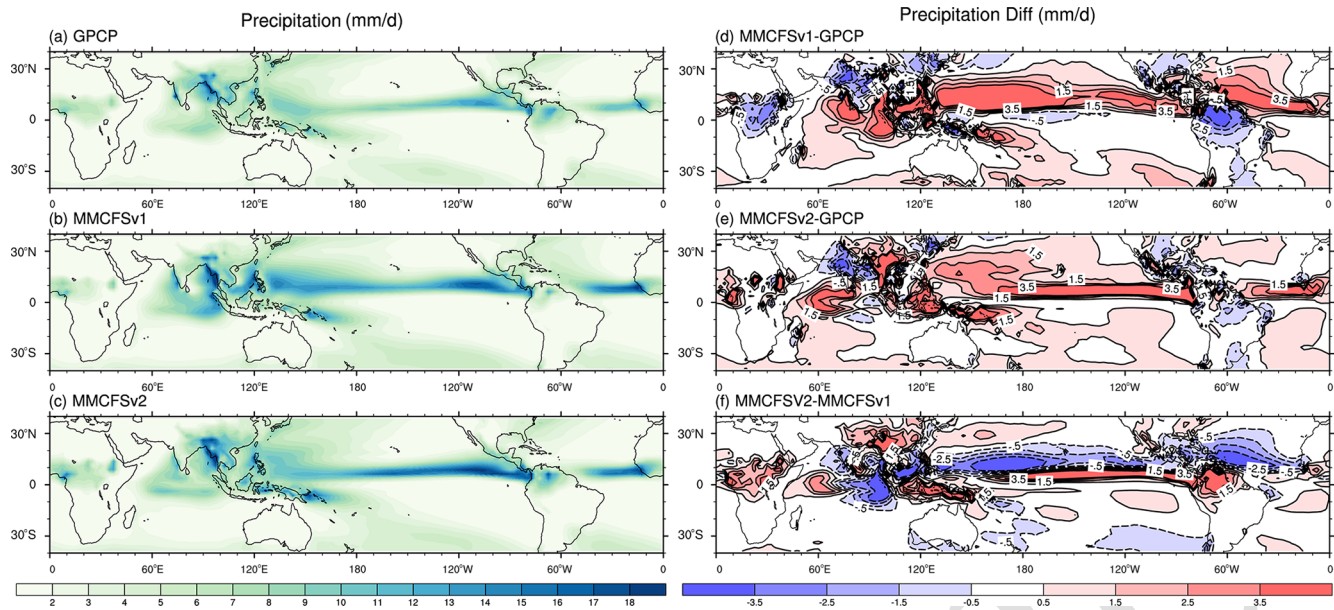

**Figure 2.** JJAS mean rainfall in **(a)** GPCP, **(b)** MMCFSv1, and **(c)** MMCFSv2, and bias in **(d)** MMCFSv1–GPCP, **(e)** MMCFSv2–GPCP, and **(f)** difference between MMCFSv1 and MMCFSv2. Dashed contours represent negative difference and solid contours are positive values.

**Table 2.** Mean ISMR, the standard deviation of ISMR, bias from observations, anomaly correlation coefficient (skill), root mean square error (RMSE) of percentage departure, and normalized (with SD) RMSE for 1998–2022. Correlation values above 99 % are shown in bold (Pearson's test).

| ISMR characteristics | | | | | | |
|---|---|---|---|---|---|---|
| Data from | Mean ($\mathrm{mm\,d^{-1}}$) | SD ($\mathrm{mm\,d^{-1}}$) | Bias ($\mathrm{mm\,d^{-1}}$) | Skill | RMSE | NRMSE |
| Observations | | | | | | |
| – GPCP | 6.99 | 0.61 | | | | |
| – IMD | 7.01 | 0.62 | | | | |
| Models | | | | | | |
| MMCFSv2 | 5.95 | 0.58 | | | | |
| – vs. IMD | | | −1.06 | **0.63** | **7.98 %** | **0.92** |
| – vs. GPCP | | | −1.04 | **0.72** | **7.01 %** | **0.82** |
| MMCFSv1 | 5.67 | 0.59 | | | | |
| – vs. IMD | | | −1.34 | **0.58** | **8.74 %** | **1.0** |
| – vs. GPCP | | | −1.32 | **0.55** | **8.99 %** | **1.06** |

ics (Fig. 3) affects the surface air temperatures over oceans, and Fig. 4 shows the warmer 2 m surface air temperature in MMCFSv2 compared with ERA5 and MMCFSv1. The cold bias of MMCFSv1 surface temperatures over the winter hemisphere (south of 15° S) has disappeared in MMCFSv2. The surface air warming is much more pronounced over the landmass.

The zonally averaged tropospheric air temperatures in Fig. 5 show both models simulating the mean observed structure consistent with observations (Fig. 5a, b, c). However, we can see that the surface warming seen in Fig. 4 produces warmer columns in the MMCFSv2 compared with both ob-

servations and MMCFSv1 (Fig. 5d, e, f). The warming, however, is confined to the summer hemisphere and MMCFSv2 is closer to observations in the southern hemisphere than MMCFSv1. The most significant upgrade from MMCFSv1 to MMCFSv2 is the ocean model. MOM6 has allowed us to use much higher ocean model resolutions than MOM4. It has also allowed the use of scale-aware parameterizations for mesoscale eddy-permitting regimes.

Figure 6 shows the difference between simulated mixed layer depth by MMCFS (v1 and v2) and the Qnet into this mixed layer. Except over the equatorial Pacific Ocean (EPO), MMCFSv1 simulates deeper mixed layer depths compared

**Table 3.** Summary of observed normal, excess, and drought years (first column uses 10 % departure from the mean and third column uses 5 % departure from the mean to define extreme years). The second column (10 %) and the fourth column (5 %) summarize hit rates and false alarms from v1 and v2 of MMCFS.

| GPCP | | 10 % Departure | GPCP | | 5 % Departure |
| --- | --- | --- | --- | --- | --- |
| Normal years 19 | V1 | False alarm 7 | Normal years 8 | V1 | False alarm 4 |
| Excess years 3 | | Hit rate 2 | Excess years 9 | | Hit rate 10 |
| Drought years 3 | V2 | False alarm 5 | Drought years 8 | V2 | False alarm 2 |
| Total extreme years 6 | | Hit rate 2 | Total extreme years 17 | | Hit rate 14 |

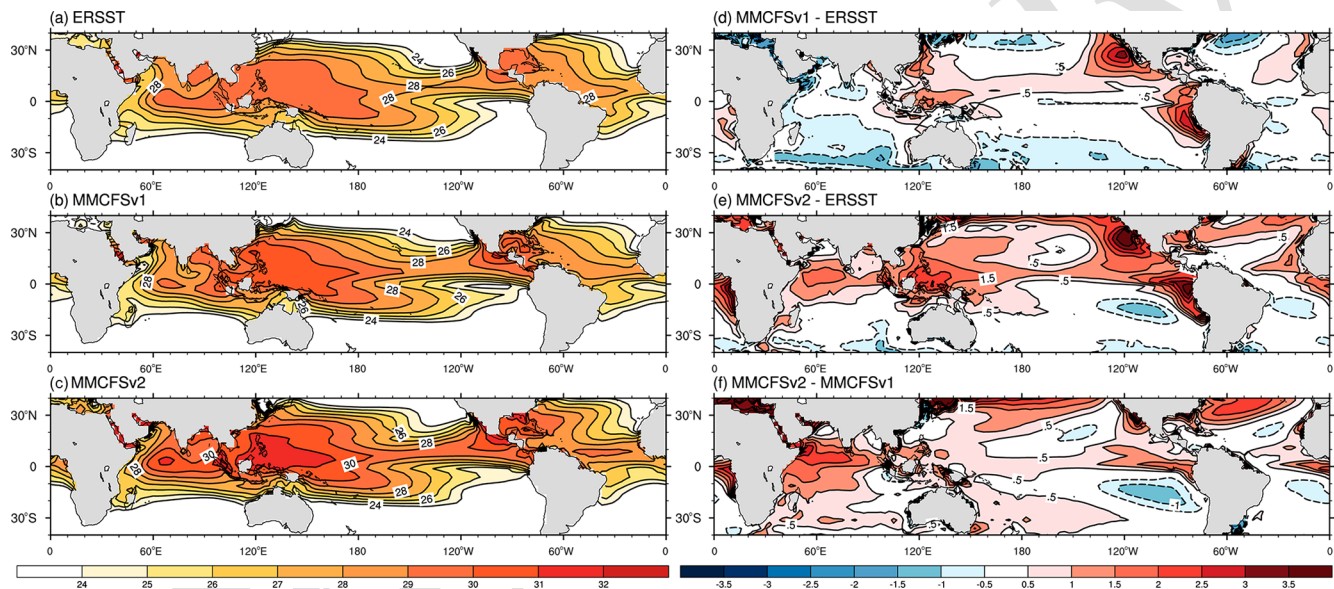

**Figure 3.** JJAS mean sea surface temperature (°C) in **(a)** ERSST (observed), **(b)** MMCFSv1, and **(c)** MMCFSv2; bias in **(d)** MMCFSv1–ERSST and **(e)** MMCFSv2–ERSST; and model difference in **(f)** MMCFSv2–MMCFSv1 over the tropics. Dashed contours highlight negative difference and solid contours are positive values.

with observations (C-GLORS). MMCFSv2 improves on this bias of MMCFSv1 with significant reduction in MLD bias (<10 m), except over the EPO, where the bias remains similar. Preliminary SST budget analysis showed that the shallower MLDs (Fig. 6) compared with MMCFS1 and similar Qnet results in warmer SSTs in MMCFSv2. Considering the fact that satellite derived SST accuracy is around 0.5° (especially AVHRR; Ahmedabadi et al., 2009), a bias of 0.5–1° cannot be considered as a significant warm bias.

### 4.1.3 Winds

JJAS mean lower tropospheric (850 hPa) observed (ERA5) and simulated (MMCFSv1 and MMCFSv2) winds in Fig. 7a–c show that both the models can capture the tropical convergence zone over the Pacific and Atlantic oceans well.

The models can also simulate the observed monsoonal circulation (Fig. 7a) over the Indian region (60–90° E) reasonably well. The difference in 850 hPa winds (Fig. 7d, e) from ERA5 shows MMCFSv2 closer to it than MMCFSv1 over most of the tropics. MMCFSv2 winds are closer to ERA5, especially over the Indian Ocean region. A significant difference in winds can be seen between the two models over the Indian Ocean region (Fig. 7f).

Focusing on the Indian region, a distinct feature of the ISM is the low-level jet (LLJ) over the Arabian Sea seen in 850 hPa winds (Fig. 7a), also popularly known as the Findlater jet (Joseph et al., 1966; Findlater 1969). Both models reproduce this low-level circulation (Fig. 7b, c). The wind bias in Fig. 7d and e shows that MMCFSv2 simulates the LLJ closer to observations than MMCFSv1. MMCFSv1 shows

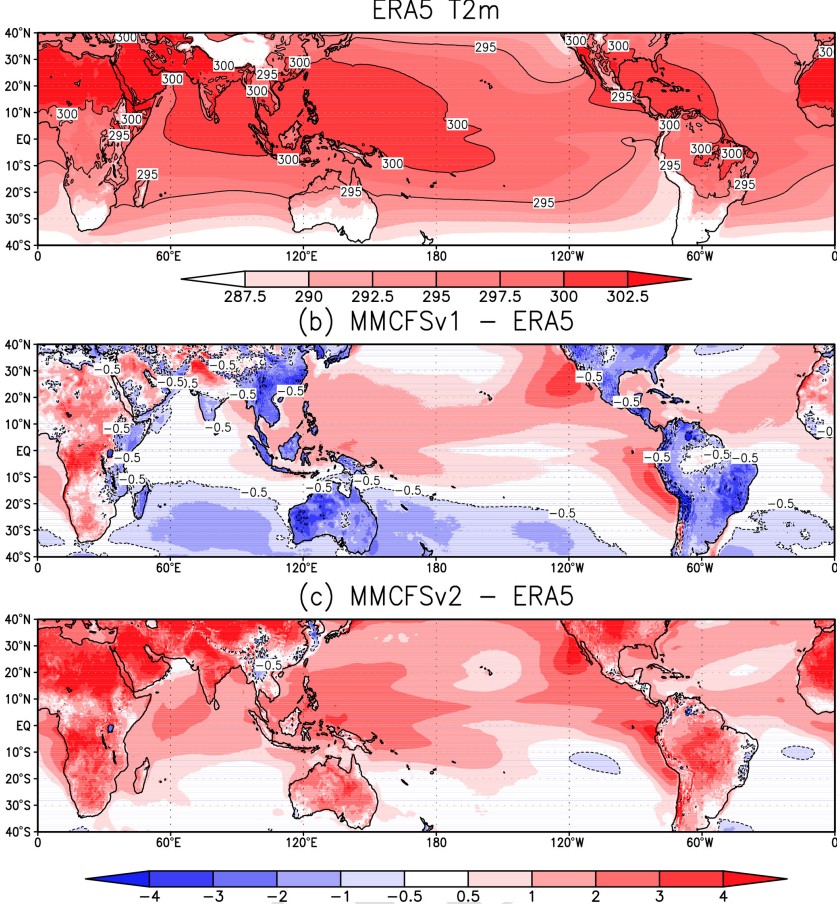

**Figure 4.** JJAS mean surface air temperature (K) at 2 m in **(a)** ERA5, and bias in **(b)** MMCFSv1–ERA5 and **(c)** MMCFSv2–ERA5. Dashed contours highlight negative difference.

strong northeast/easterly wind bias over the southern Indian region, the Arabian Sea, and the Bay of Bengal. Significant westerly wind bias is seen in MMCFSv1 over the entire southern and equatorial Indian Ocean (Fig. 7). Figure 7d and e show both models having a low-level anti-cyclonic circulation bias over the Indian subcontinent. Compared with MMCFSv1, the low-level anti-cyclonic circulation bias is significantly reduced in MMCFSv2 (Fig. 7e, f). MMCFSv1 simulates stronger northeasterly or easterly wind bias compared with MMCFSv2 (Fig. 7f) over the equatorial region. This may be due to the enhanced convection in the eastern equatorial Indian Ocean in MMCFSv2 (Fig. 2).

Both models simulate the observed upper tropospheric tropical divergence and subtropical jets (in 200 hPa winds; Fig. 8a, b, c). The monsoonal circulation over the Indian region is also evident in both models. The difference in 200 hPa winds (Fig. 8d, e) shows MMCFSv1 winds are closer to ERA5 over the Indian oceanic region (15° S–15° N), and MMCFSv2 winds are closer to ERA5 over the Asian landmass (north of 15° N). MMCFSv2 simulates a weaker (stronger) subtropical jet over the northern (southern) Hemisphere compared with MMCFSv1 and is closer

to ERA5. Significant differences in 200 hPa winds can be seen between the two models over the Indian landmass, Indian Ocean, southern Pacific, and Atlantic oceanic regions. The mean upper tropospheric (200 hPa) winds during ISM are characterized by the tropical easterly jet (TEJ) and Tibetan anticyclone (Fig. 8a) (Krishnamurti et al., 1976). Both models can get these upper tropospheric circulation features (Fig. 8b, c). Compared with MMCFSv1, MMCFSv2 has a weaker westerly bias over India (Fig. 8d, e, f).

Figure 9a shows the longitudinally (global) averaged zonal ($U$) winds from ERA5. The models capture the easterly jet in the tropical convergence zone and the westerly jets in the mid-latitudes (Fig. 9b, c). MMCFSv1 simulated wind bias (Fig. 9d) shows reduced strength of easterlies in the southern tropics and westerlies in the southern mid-latitudes. MMCFSv2 shows (Fig. 9e) a weaker strength in the mid and upper tropospheric region of the tropical easterly jet. A weaker westerly jet in MMCFSv2 can be seen at both 50–60° N and 60° S. The winds in the northern hemisphere are close to ERA5 in MMCFSv1. The tropical surface zonal winds close to ERA5 in MMCFSv2.

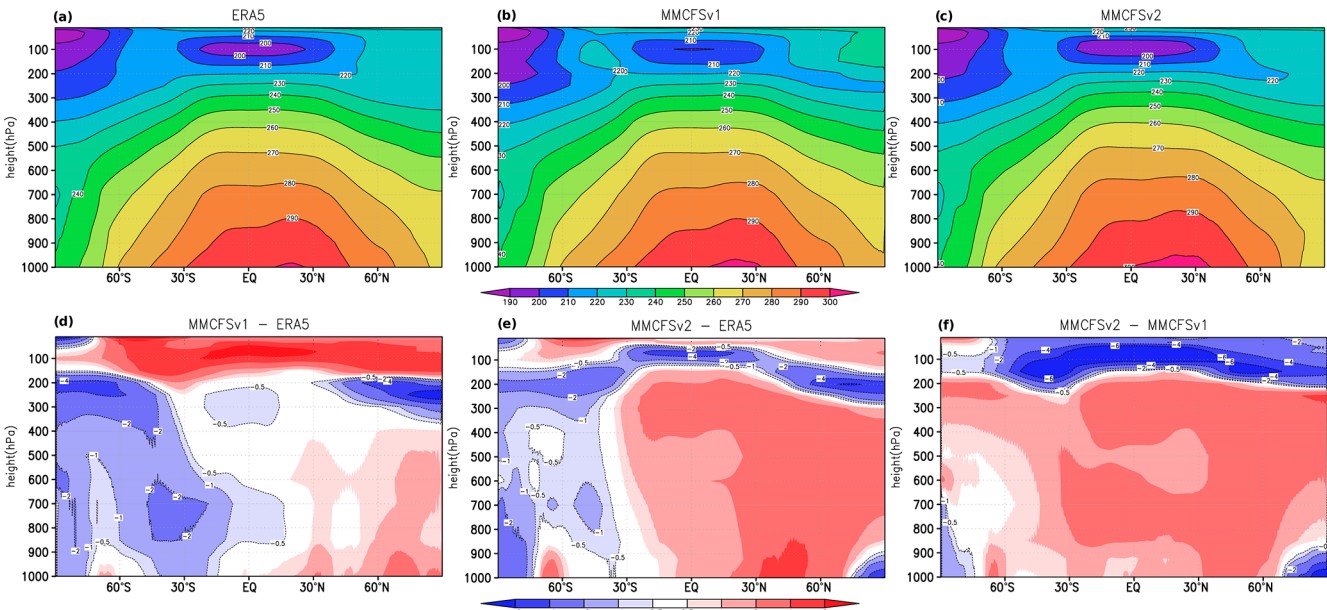

**Figure 5.** JJAS zonal mean temperature (K) in **(a)** ERA5, **(b)** MMCFSv1, and **(c)** MMCFSv2; bias in **(d)** MMCFSv1–ERA5 and **(e)** MMCFSv2–ERA5; and **(f)** shows the difference between MMCFSv1 and MMCFSv2. Dashed contours highlight negative difference.

Comparing the two simulated winds with each other (Fig. 9f), we see a slightly reduced strength of the upper-level westerly jet in the summer hemisphere (45° N, 200–500 hPa in Fig. 9f) and increased strength in the winter hemisphere (45° S, 200–900 hPa) in MMCFSv2 compared with MM-CFSv1. Overall, both models simulate the zonal mean tropical winds reasonably well, with slightly different strengths of tropical and sub-tropical jets. Since ISM is significantly affected by ENSO through the Hadley cell, we expect significantly different teleconnection patterns between ISMR and ENSO in the two models (Fig. 14; discussed later). This also encourages us to look at the wind shear structure simulated by the two models.

### 4.1.4 Wind shear

The vertical wind shear over the Asian summer monsoon (ASM) region plays an important role in modulating the northward propagation of monsoon intraseasonal oscillations (MISO) (Jiang et al., 2004). Figure 10 shows the observed and the model-simulated JJAS seasonal mean of easterly wind shear. The wind shear is computed as the difference between 850 and 200 hPa zonal ($U$) winds. Large positive wind shear (greater than $12\,\mathrm{m\,s^{-1}}$) is observed (Fig. 10a) over the southern Asian region during the monsoon season. Positive wind shear is also seen over the Sub-Saharan region, the Indian Ocean, the western and eastern Pacific, and equatorial Atlantic regions. Negative wind shear is observed in the central North Pacific, South Pacific, and North Atlantic oceanic regions. Both models capture these features well (Fig. 10b, c). The wind shear bias (Fig. 10d, e) shows that MMCFSv2

shear is closer to that of ERA5 (difference less than $5\,\mathrm{m\,s^{-1}}$) compared with MMCFSv1 (difference greater than $5\,\mathrm{m\,s^{-1}}$) over most of the tropical regions. MMCFSv1 largely simulates a high negative bias over the Northern Hemisphere and positive bias over the Southern Hemisphere (Fig. 10d) which has improved significantly in MMCFSv2 (Fig. 10e). The bias in MMCFSv2 is significantly lower compared with that of MMCFSv1 over Asian and African landmasses and most of the Pacific and Atlantic oceans.

ERA5 reanalysis shows a positive wind shear over the ASM domain (Fig. 10a). The wind shear over the ASM region is underestimated in MMCFSv1 and MMCFSv2 (Fig. 10b, c), consistent with the weak monsoon winds and TEJ, as seen in Figs. 7 and 8. However, there is a considerable difference between the two models: whereas MMCFSv1 produces a large negative bias over Indian land and a positive bias over the southern Indian Ocean compared with observations (Fig. 10d), MMCFSv2 bias is positive over Indian land and predominantly negative over the Indian oceanic region. This difference between the simulations is much clearer in Fig. 10f. Therefore, there will be a considerable difference between the northward propagation speeds of the MISOs of MMCFSv1 and MMCFSv2. (The difference in MISO characteristics will be explored in greater detail in a future study.) MMCFSv2 underestimates shear in the western equatorial Indian Ocean/Arabian Sea, primarily due to a simulated weak easterly jet at 200 hPa (Fig. 8).

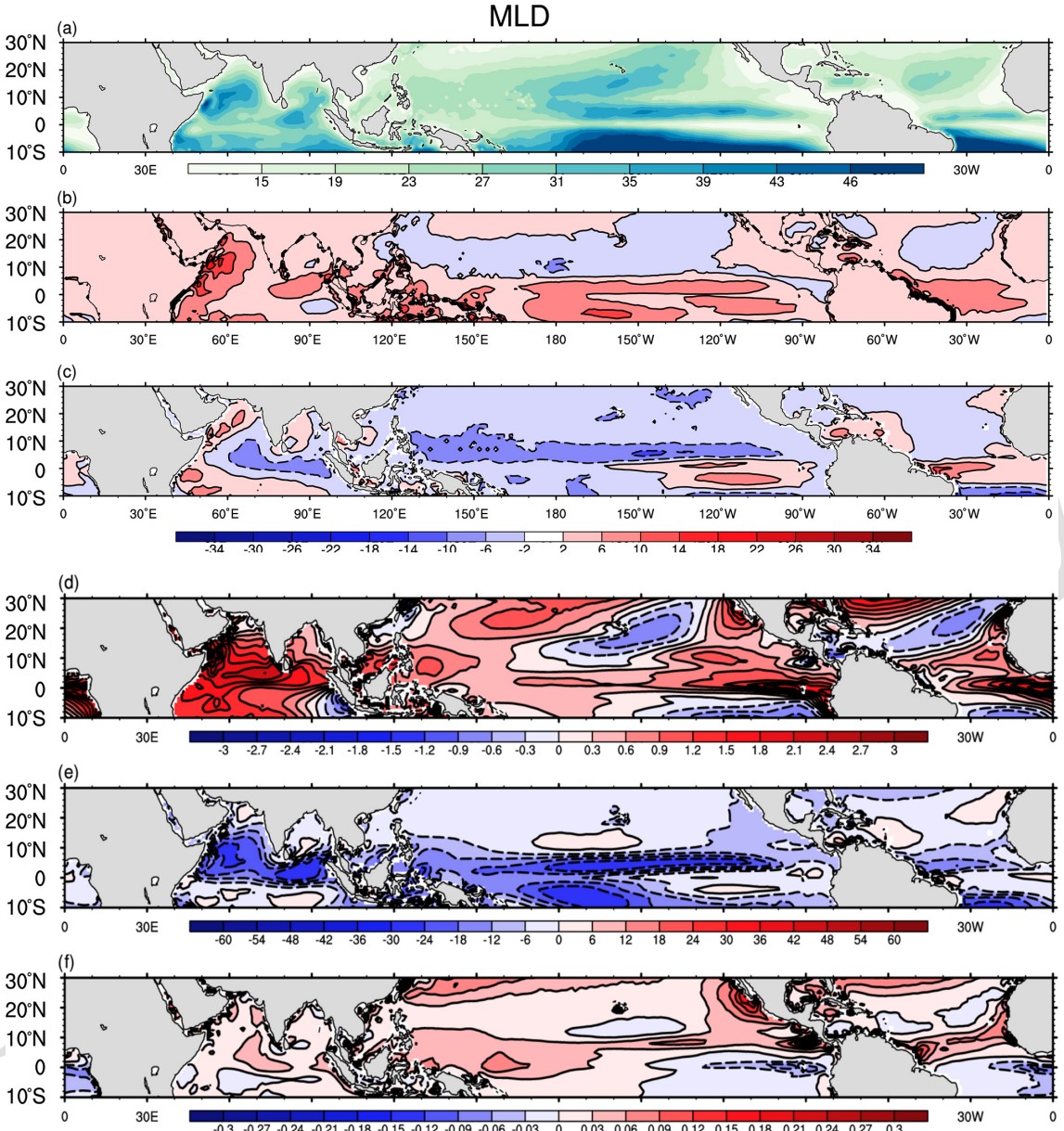

**Figure 6.** Panel **(a)** shows mixed layer depth from C-GLORS, and **(b)** and **(c)** show model bias of MLD from C_GLORS. Panel **(d)** shows the SST difference between v2 and v1 (v2–v1). Panel **(e)** shows the MLD difference (v2–v1) and **(f)** shows the Qnet contribution to MLD. Dashed contours highlight negative values.

### 4.1.5 Interannual variability of ISMR and potential skill

The year-to-year variations of the area-averaged JJAS rainfall over the Indian land region are shown in Fig. 11 TS12. The ob-
served all-India summer monsoon rainfall time series is pre-
pared from the India Meteorological Department (IMD) grid-
ded land rainfall and GPCP rainfall data. Figure 10 shows
that both models can capture the recent rainfall deficit years
of 2014 and 2015, as can be seen from GPCP. Out of 25 re-
forecast years, MMCFSv2 could capture 20 years correctly,

while MMCFSv1 could capture 15 years. Failure of the hind-
casts in 2019 and 2000 is required to be analyzed in detail. A
detailed analysis is required to understand the performance
of the MMCFSv2.

Table 2 summarizes the model skill in reproducing interan-
nual variability of observed ISMR during 1998–2022. MM-
CFSv2 shows improvements in producing the mean of JJAS
rainfall over MMCFSv1 by reducing the dry bias from 1.32
to 1.04 mm d$^{-1}$ ($\sim 4\%$) with respect to GPCP. MMCFSv2
captures the phase of interannual variability with a higher
skill of 0.72 over 0.55 of MMCFSv1 when GPCP is consid-

## 850 hPa Winds

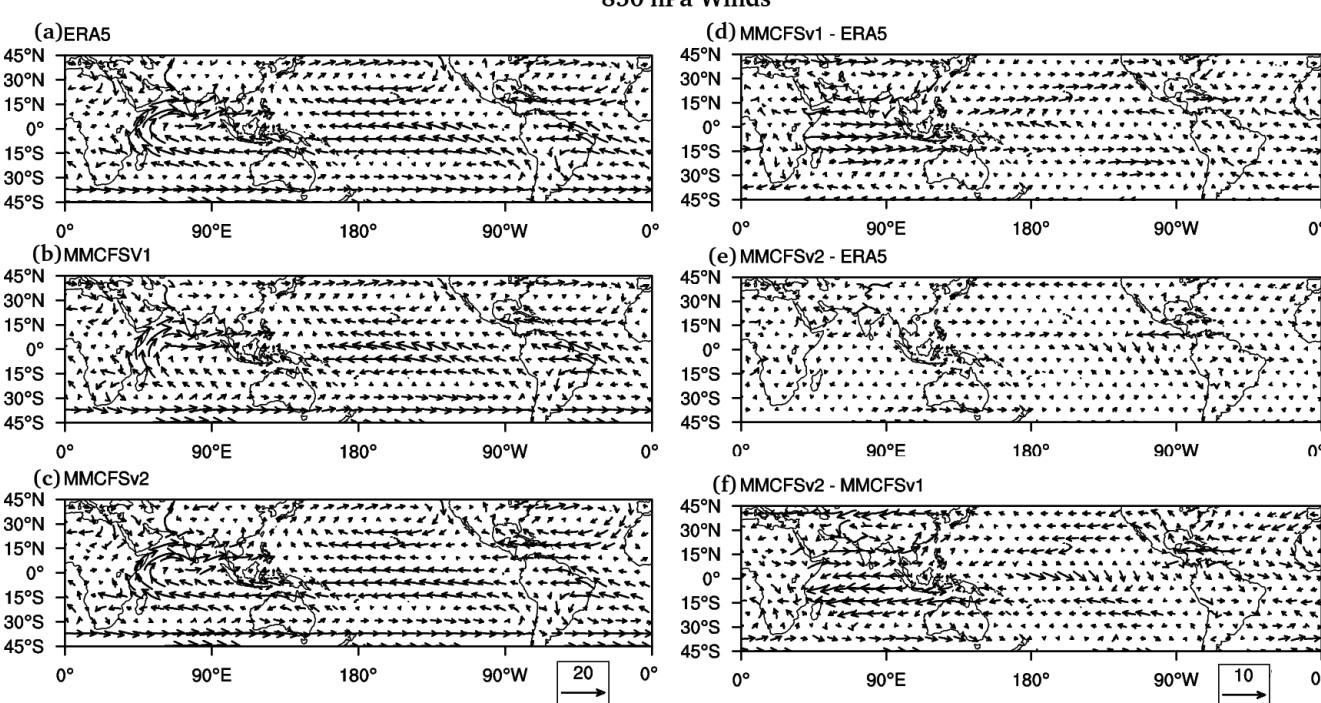

**Figure 7.** JJAS climatological mean winds (m s$^{-1}$) at 850 hPa in **(a)** ERA5, **(b)** MMCFSv1, and **(c)** MMCFSv2; bias in **(d)** MMCFSv1-ERA5 and **(e)** MMCFSv2-ERA5; and **(f)** shows the difference between MMCFSv2 and MMCFSv1.

## 200 hPa Winds

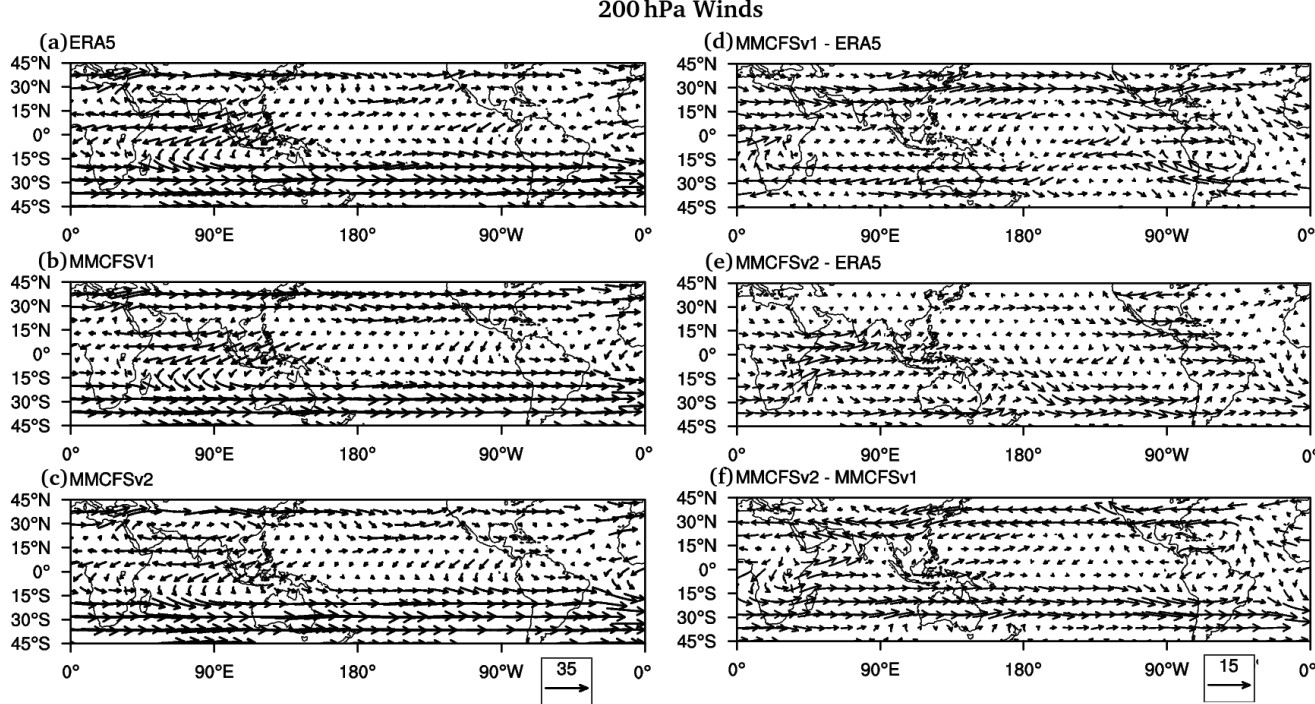

**Figure 8.** JJAS climatological mean winds (m s$^{-1}$) 200 hPa in **(a)** ERA5, **(b)** MMCFSv1, and **(c)** MMCFSv2; bias in **(d)** MMCFSv1–ERA5 and **(e)** MMCFSv2-ERA5; and **(f)** shows the difference between MMCFSv2 and MMCFSv1.

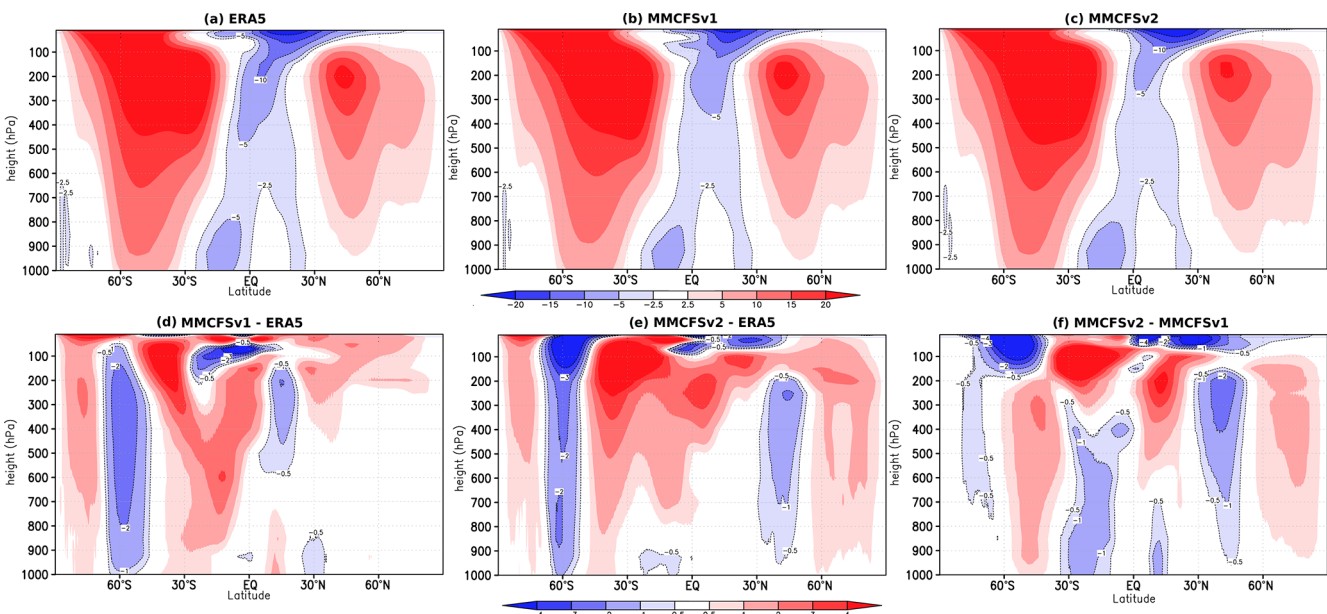

**Figure 9.** JJAS climatological zonal mean wind $(\mathrm{m\,s^{-1}})$ in **(a)** ERA5, **(b)** MMCFSv1, and **(c)** MMCFSv2; bias in **(d)** MMCFSv1–ERA5 and **(e)** MMCFSv2-ERA5; and **(f)** shows the difference between MMCFSv1 and MMCFSv2. Negative values are highlighted with contour lines.

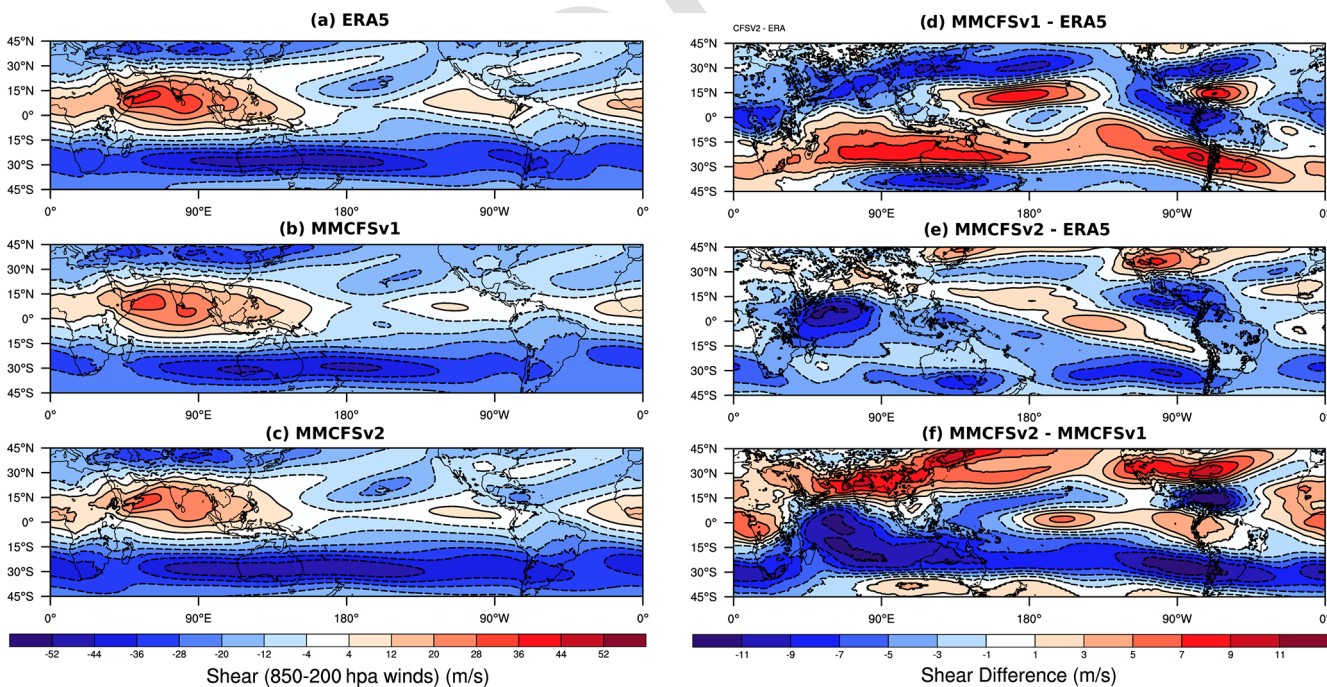

**Figure 10.** JJAS seasonal mean easterly wind shear (U850–U200; $\mathrm{m\,s^{-1}}$) in **(a)** observations (ERA5), **(b)** MMCFSv1, and **(c)** MMCFSv2. Seasonal mean easterly wind shear biases (model–observation) in **(d)** MMCFSv1 and **(e)** MMCFSv2. Panel **(f)** shows the difference in simulated seasonal mean easterly wind shear between MMCFSv2 and MMCFSv1 hindcast runs. Dashed contours highlight negative values and solid contours are positive values.

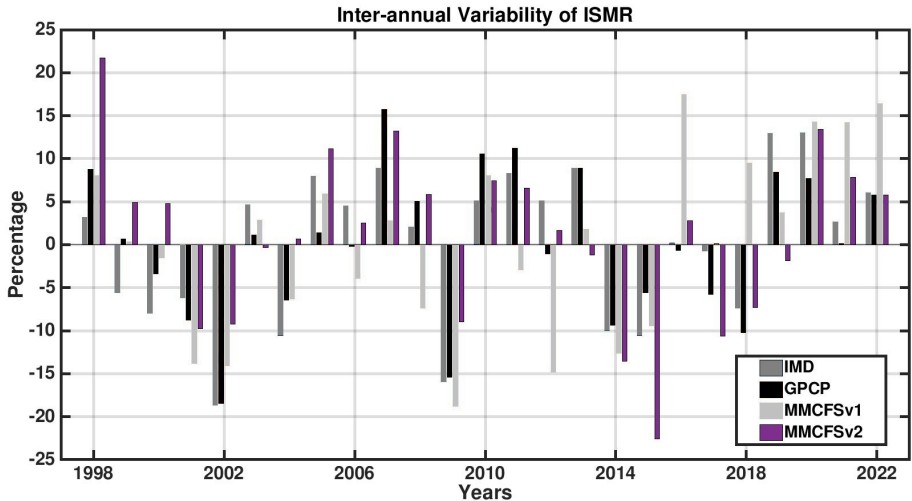

**Figure 11.** Interannual variability of area-averaged rainfall over the Indian landmass from model hindcasts (MMCFSv2 and MMCFSv1) and two observational datasets (IMD and GPCP). TS11

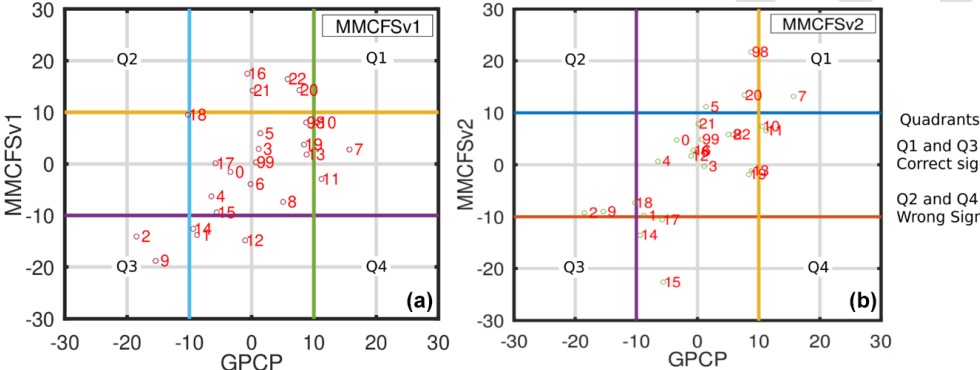

**Figure 12.** Scatter plot of ISMR anomaly (percentage) from GPCP (*x* axis) and MMCFS (*y* axis). Panel **(a)** is MMCFSv1 and **(b)** is MMCFSv2.

ered as observation. Hence, MMCFSv2 improves the phase skill by 17 %.

Figure 12 shows the scatter plot of the observed ISMR anomaly (expressed as percentage departures from mean) from GPCP and MMCFS. From the scatter plot it is evident that many observed normal years were predicted as extremes in MMCFSv1. Hence, we calculated the false alarm rates and the hit rates for both the models. We used two criteria for defining normal years, viz. 10 % and 5 % departure from the climatological mean. Table 3 TS13 summarizes the false alarms and hit rates. As seen from the table, MMCFSv1 has a higher false alarm rate and a lower hit rate than MMCFSv2.

Pillai et al. (2018a) compared the seasonal prediction skill of ISMR in MMCFSv1 (T382) with the US National Multi-Model Ensemble (NMME) project for the simulation years of 1981–2009. They found that MMCFSv1 has better skill in reproducing interannual variability of ISMR (ACC = 0.55) compared with the other NMME models (ACC <0.4) and MMCFSv1 is better at simulating the observed standard

deviation of ISMR. The Taylor diagram (Taylor, 2001) in Fig. 13 shows the skill of MMCFS (v1 and v2) and NMME models in reproducing observed standard deviation (SD) and normalized root mean square error normalized (NRMSE). SD and root mean square error are normalized with observed standard deviation. Of these NMME models, GFDL_FLORA, GFDL_FLORB, and SPISV2 have data for the years of 1998–2021. We found that removal of year 2022 from other models does not change the scores significantly. There are five models which simulate the observed SD reasonably well (normalized SD approximately 1.0), viz. MM-CFSv2, GFDL_Aero, SIPSv2, SPSIC3, and GMAO. All the other models have larger or smaller standard deviations with respect to observations. A 10 % deviation from the climatological mean is sufficient to have an excess or a drought monsoon over India (Singh et al., 2015). Hence, getting the NRMSE below 1.0 is crucial. Two models which stand out in terms of NRMSE are MMCFSv2 (0.82) and GFDL_Aero (0.85). All the other models simulate the NRMSE larger than

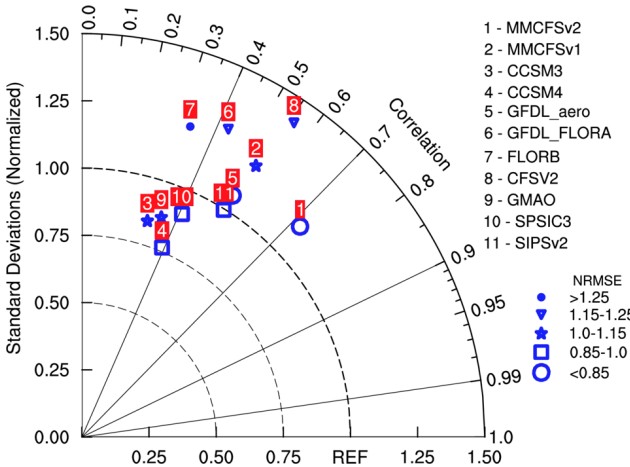

**Figure 13.** Taylor diagram showing the normalized RMSE pattern correlation coefficients and normalized standard deviation of the JJAS mean ISMR of the MMCFS and NMME models with respect to GPCP observations. The NMME model simulation period is 1998–2021 and the MMCFS period is 1998–2022.

0.85. MMCFSv2 reduces the NRMSE from 1.04 of MM-CFSv1 to 0.82 (which is about 20 %) with respect to GPCP. GFDL_Aero also has a lower ACC of 0.53 compared with 0.72 of MMCFSv2. MMCFSv2 has the highest skill in capturing the interannual variability of ISMR compared with all the other models. Hence, in terms of SD, NRMSE, and the ACC, MMCFSv2 stands out compared with all the other NMME models and MMCFSv1.

The uncertainty in initial conditions is inevitable due to gaps in observational networks and the limitations of data assimilation systems. Therefore, it is not possible to know the "true" state of the earth system, which serves as a starting point for the seasonal simulations. Ensemble forecasting techniques (such as the one used in this study) are employed to account for the initial state's uncertainty. If we assume the model is perfect, the uncertainty in initial conditions puts an upper limit to predictability. This upper limit is termed as the potential predictability and estimates the maximum skill the "perfect" model can achieve. Let us say that the forecast for the variable "$x$" using the initial condition "$i$" has a probability distribution $P(x|i)$. This forecast reaches an equilibrium state asymptotically with the distribution $q(x)$. The distance between these two distributions is a measure of predictability and is termed as the relative entropy (RE) or the Kullback–Leibler distance. If this forecast distribution is identical to the climatological distribution, there is no predictability. RE can be estimated using the following equation, following Kleeman (2002), under the assumption that both the distributions are Gaussian:

$$RE = \frac{1}{2}\left[\ln\left(\frac{\sigma_x^2}{\sigma_{x|i}^2}\right) + \frac{\sigma_{x|i}^2}{\sigma_x^2} + \frac{(\mu_{x|i} - \mu_x)^2}{\sigma_x^2} - 1\right], \quad (1)$$

where $\sigma_{x|i}^2$ and $\sigma_x^2$ are the ensemble (forecast) and climatological variance, respectively. $\mu_x$ and $\mu_{x|i}$ are the climatological and ensemble mean, respectively. Climatological variance is estimated as the sum of signal and noise variance (DelSole and Tippett, 2007) as

$$\sigma_x^2 = \frac{1}{N}\sum_{i=1}^{N}\sigma_{x|i}^2 + \frac{1}{N}\sum_{i=1}^{N}\left(\mu_{x|i} - \mu_x\right)^2. \quad (2)$$

The average of RE across all ensembles is the mutual information (MI). Potential skill (PS) is defined as

$$PS = \sqrt{1 - e^{(-2\,MI)}}. \quad (3)$$

The actual skill achieved by the model in this paper is computed using the anomaly correlation coefficient. The PS for MMCFSv2 is 0.79 using the above expression, while the actual skill obtained is 0.72 (Table 2). The PS and actual skill for MMCFSv1 for 1981–2017 is 0.72 and 0.38, respectively (Pillai et al., 2018b). This indicates that the actual model skill of MMCFSv2 is very close to the perfect model skill. Further improvements to the individual model components shall bring the actual skill closer to the potential skill.

Recent studies (Ramu et al., 2016; George et al., 2016; Pillai et al., 2022 TS14) have shown that the seasonal prediction skill of monsoon in MMCFSv1 is significantly impacted by the El Niño–Southern Oscillation (ENSO)–monsoon relationship. MMCFSv1 also has some limitations in representing the relationship between Indian Ocean SST and monsoon. We therefore now analyze the simulated teleconnections of the observed and simulated ISMR with different oceanic regions across the world.

## 4.2 Teleconnections

Earlier studies have found that the year-to-year variability of ISMR is mainly linked to the Pacific ENSO and Indian Ocean dipole (IOD) (Webster et al., 1992; Kumar et al., 1999b; Saji et al., 1999; Ashok et al., 2004; Rajeevan and Francis, 2007; Rajeevan and Pai, 2007). Atlantic zonal and meridional modes (AZM and AMM) also play a role in modulating ISMR. AZM is the oscillatory normal mode (zonal) seen in the principal oscillation pattern analysis of SST (Ding et al., 2010; Zebiak, 1993), while AMM refers to the leading maximum covariance analysis mode in the tropical Atlantic. IOD is seesaw in sea surface temperature anomalies between the western and eastern equatorial Indian Ocean.

Recently, Sabeerali et al. (2019) explored the impact of the Atlantic zonal mode on ISM at interannual timescales in recent years using CFSv2. Here, we compare the ability of the models to simulate the teleconnections between ISM and ENSO, IOD, and the Atlantic modes. Table 4 summarizes the skill of models in simulating the oceanic modes (ENSO, eastern Indian Ocean dipole (EIOD), AMM, and AZM) and their teleconnections with ISMR. Here, only the eastern pole of the IOD is considered, as it is the stationary part of the IOD

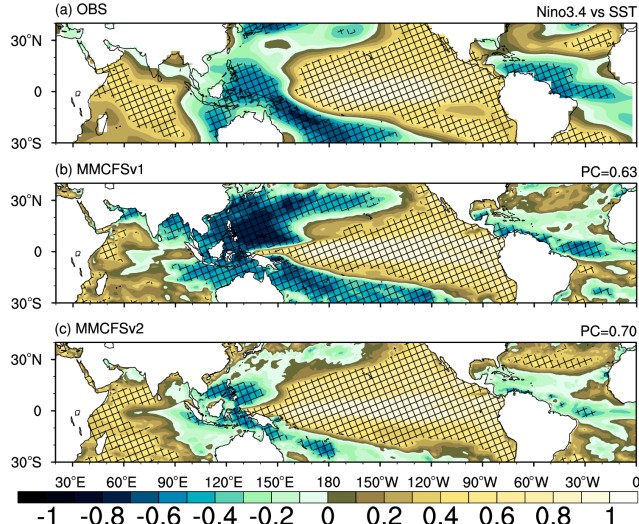

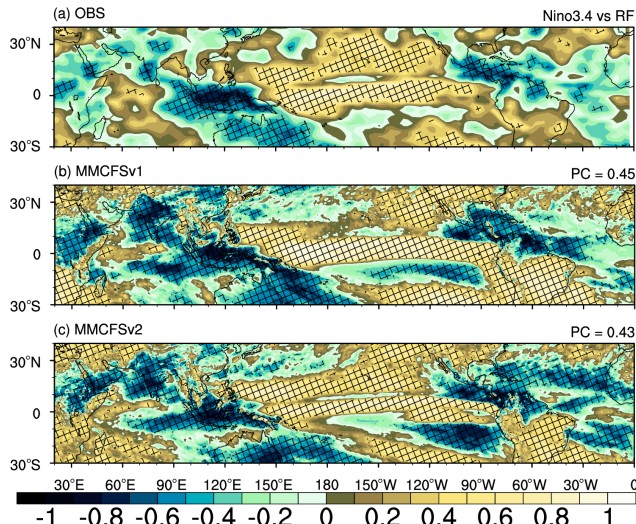

**Figure 14.** Correlation between JJAS Niño 3.4 (index) SST anomalies and tropical SST anomalies in **(a)** observations (ERSST), **(b)** MMCFSv1, and **(c)** MMCFSv2. The hatching shows statistical significance at the 95 % confidence level.

**Figure 15.** Correlation between JJAS Niño 3.4 (index) SST anomalies and tropical rainfall anomalies in **(a)** observations (GPCP), **(b)** MMCFSv1, and **(c)** MMCFSv2. The hatching shows statistical significance at the 95 % confidence level.

(Rao et al., 2009). Niño 3.4 SST anomalies are averaged over the region of 170–120° W and 5° S to 5° N. Atlantic meridional mode is the SST anomaly difference between northern (5–15° N, 50 to 20° W) and southern (5–15° S, 20° W to 10° E) points, and AZM is the SST anomaly over 5° S to 3° N, 20° W to 10° E.

### 4.2.1 ENSO

Models can capture Niño 3.4 with high skill (Table 4). The spatial distribution of simultaneous correlations between Niño 3.4 SST anomalies (index) and tropical SST anomalies in JJAS are shown in Fig. 14. Positive correlations over the eastern/central tropical Pacific and western/central Indian Ocean are observed. Moreover, negative correlations are observed over the western tropical Pacific, eastern equatorial Indian Ocean, and tropical Atlantic Ocean (Fig. 14a). MMCFSv2 simulates these large-scale teleconnection patterns associated with Niño 3.4 over the tropics with a higher pattern correlation of 0.70 than MMCFSv1 (PC = 0.63) (Fig. 14b, c). In MMCFSv1, positive correlations over the Pacific and western Indian oceans are weaker than observations. MMCFSv2, on the other hand, captures these teleconnection patterns in the tropical Indian Ocean and over the Pacific regions reasonably well; hence, pattern correlation is higher for MMCFv2.

The spatial plot of the correlation between the boreal summer Niño 3.4 anomaly index and rainfall anomaly over the tropical region is shown in Fig. 15. Observations show that the Niño 3.4 SST anomalies are negatively correlated (correlation coefficient of −0.64; Fig. 15a; Table 4) with rainfall over the Indian land region (Fig. 15a). Consistent with ob-

servations, both MMCFSv1 and MMCFSv2 simulate this inverse relationship reasonably well, albeit with an overestimation. The Niño 3.4 and ISMR teleconnection in MMCFSv2 (−0.75) is closer to observations (−0.64) than in MMCFSv1 (−0.83). Additionally, observations show a strong positive correlation between the Niño 3.4 SST anomalies and rainfall over the tropical Pacific. MMCFSv2 can and MMCFSv1 cannot simulate this positive correlation over the North Pacific region (Fig. 15b, c). A moderate negative correlation is seen over the Atlantic Ocean (Fig. 15a), which is better captured by MMCFSv1. Except over the southeastern equatorial Pacific and Atlantic oceans, MMCFSv2 can reproduce the Niño 3.4-induced rainfall pattern over the Bay of Bengal region and the northern and equatorial Pacific. Both models can capture the correlations over the Indian Ocean, with a slightly overestimated Niño 3.4-induced rainfall pattern (Fig. 15b, c).

### 4.2.2 EIOD and other tropical modes

MMCFSv1 has a higher skill of 0.58 in capturing EIOD than MMCFSv2 (0.42 as shown in Table 4). The spatial pattern of correlation between ERSST over the EIOD box (10° S to the Equator, 90–110° E) and tropical SST anomalies during JJAS season is shown in Fig. 16. Here, only the eastern pole of the IOD is considered, as it is the stationary part of the IOD (Rao et al., 2009). Observations show a strong positive correlation between the Indo-Pacific warm pool region and the equatorial Atlantic Ocean. Negative correlations exist between the tropical Pacific Ocean and the western tropical Indian Ocean (Fig. 16a). The pattern correlation of this teleconnection has improved from 0.31 in MMCFSv1 to 0.38 in MMCFSv2. Both models capture the positive correlations

**Table 4.** Teleconnections of ISMR with different oceanic indices and skill of the models in capturing these modes (95 % statistically significant values in bold; Pearson's test).

| Teleconnection (with ISMR) | Niño 3.4 | EIOD | AMM | AZM |
|---|---|---|---|---|
| – Observations | **−0.64** | −0.04 | 0.18 | 0.19 |
| – MMCFSv2 | **−0.75** | 0.33 | −0.07 | **0.46** |
| – MMCFSv1 | **−0.83** | **0.68** | 0.35 | 0.08 |
| Skill | Niño 3.4 | EIO | AMM | AZM |
| – MMCFSv2 | **0.83** | **0.42** | 0.15 | 0.32 |
| – MMCFSv1 | **0.82** | **0.58** | 0.01 | 0.13 |

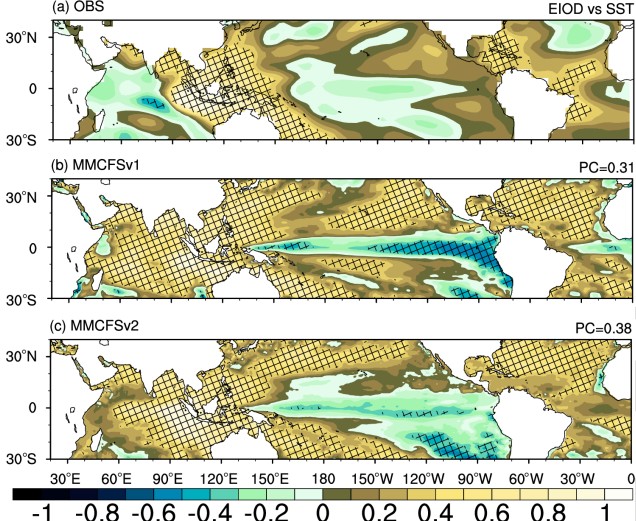

**Figure 16.** Correlation between SST over the eastern IOD box (10° S to Equator, 90–110° E) and tropical SST anomalies during JJAS. Panel **(a)** shows observations (ERSST), **(b)** MMCFSv1, and **(c)** MMCFSv2. The hatching shows statistical significance at the 95 % confidence level.

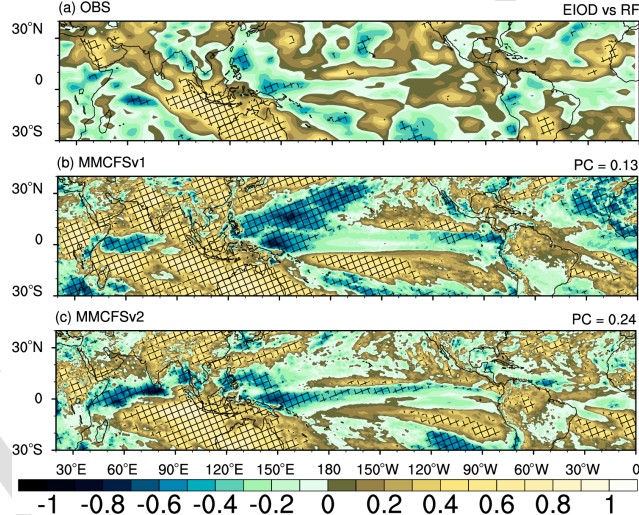

**Figure 17.** Correlation between the SST anomalies over the eastern equatorial Indian Ocean (index) and rainfall anomalies over the tropical regions. Panel **(a)** shows observations (GPCP), **(b)** MMCFSv1, and **(c)** MMCFSv2. The hatching shows statistical significance at the 95 % confidence level.

over the Indo-Pacific warm pool region (Fig. 3b, c). Furthermore, both models simulate a basin-wide positive correlation over the entire Indian Ocean, with weaker positive or insignificant correlations in the western Indian Ocean in MMCFv2 (Fig. 16b, c), in contrast to the observed negative correlation pattern (Fig. 16a).

Figure 17 shows the spatial map of the correlation between the SST anomalies over the eastern equatorial Indian Ocean and rainfall anomalies over the tropical region. A positive correlation (not significant) in most parts of south/central India is observed. An expected strong positive correlation exists over the eastern Indian Ocean and northern Australia. Pacific and Atlantic oceanic rainfall has a weak correlation with eastern equatorial Indian Ocean SST anomalies (Fig. 17). MMCFSv2 simulates this EIOD-induced rainfall pattern over the central and southern Indian regions. It is, however, the opposite of the observed relation over the northern Indian Ocean region (Fig. 17b). MMCFSv1 over-

estimates this positive correlation over the Indian region compared with MMCFSv2. The pattern correlation between these teleconnections (EIOD SST–rainfall; Fig. 17) has improved from 0.13 in MMCFSv1 to 0.24 in MMCFSv2. The observed teleconnection between ISMR and EIOD is −0.04 (Table 4). On the contrary, MMCFSv2 and MMCFSv1 show a strong positive teleconnection relationship between ISMR and IOD of 0.33 and 0.68, respectively (Table 4). The strong unrealistic in-phase relation between ISMR and EIOD is significantly reduced in MMCFSv2 from 0.68 to 0.33.

Figure 18 shows the simultaneous correlation between the JJAS ISMR anomaly index and tropical SST anomalies. Observed ISMR correlates significantly (negatively) with SST anomalies over the central North Pacific (around 0–20° N, 150–240° E). The correlation is weaker and positive over the northwestern Pacific region. ISMR is significantly (positively) correlated with SST anomalies over the North Atlantic region and is weakly correlated with Indian Ocean SST

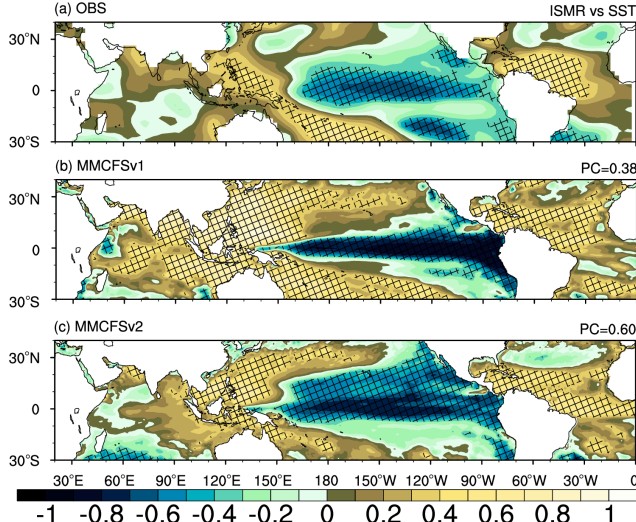

**Figure 18.** Correlation between ISMR and global SST anomalies. Panel **(a)** shows observations (ERSST), **(b)** MMCFSv1, and **(c)** MMCFSv2. The hatching shows statistical significance at the 95 % confidence level.

anomalies (observations in Fig. 18a). Both models overestimate this correlation over the western North Pacific region compared with observations. MMCFSv1 shows stronger teleconnections between the ISMR and Pacific Ocean compared with MMCFSv2 (stronger negatives over the central and eastern Pacific, and stronger positives over the western Pacific). North Atlantic SST anomalies are well captured by both models (Fig. 18b, c). MMCFSv2 can reproduce the observed correlation over the Indian and Pacific oceans much better than MMCFSv1. Overall, the pattern correlation between the MMCFSv2 and observed teleconnection is much higher (at 0.60) than that of MMCFSv1 (at 0.38). We further assessed the model in simulating the teleconnection between Atlantic meridional and zonal modes and ISMR, and found that MMCFSv2 cannot simulate the observed AZM and AMM teleconnections (Table 4).

## 5 Summary and discussion

A new Monsoon Mission Coupled Forecast System version 2 (MMCFSv2) model has been deployed at IITM to replace the currently operational MMCFSv1. MMCFSv2 brings a substantial number of component upgrades over MMCFSv1. These upgrades include the use of the MOM6 ocean model over MOM4, the CICE5 model over the SIS sea-ice model of MMCFSv1, and the semi-Lagrangian dynamical core for integrating the GFS atmospheric model over the Eulerian model. The coupler in the MMCFSv1 is based on the NEMS framework. This framework allows the model to interface with numerous external model components and brings in much-needed modularity for easy future upgradability. Cou-

pled hindcast simulations with April initial conditions from CFSR have been carried out for 25 years (from 1998 to 2022). This dataset will be the baseline for future sensitivity studies using MMCFSv2.

We documented the MMCFSv2 model skill (compared with MMCFSv1) in simulating mean tropical SST, precipitation, and circulation. We also documented the skills in simulating Indian summer monsoon at seasonal timescales, as well as mean and interannual variability of ISMR and its teleconnections with ENSO and IOD, AMM, and AZM. MMCFSv2 captures all the large-scale features during the JJAS season reasonably well. It shows improvements in many large-scale meteorological features over MMCFSv1. The wet rainfall bias over the North Pacific is reduced considerably in MMCFSv2 compared with MMCFSv1. The wind shear bias is reduced considerably in MMCFSv2. Lower tropospheric winds are much better simulated in MMCFSv2 compared with MMCFSv1. One of the biggest weaknesses of most climate models in simulating the Indian monsoon is the dry bias compared with observations. MMCFSv2 reduced this bias compared with MMCFSv1. MMCFSv2 simulates upper and lower tropospheric winds much better. Wind shear is also much closer to observations over Indian landmass in MMCFSv2 compared with MMCFSv1.

MMCFSv2 showed improvements in reproducing the mean of JJAS rainfall over MMCFSv1 by reducing the bias from 1.32 to 1.04 ($\sim 4\,\%$) with respect to GPCP. MMCFSv2 captured the observed (GPCP) phase of interannual variability with a higher skill of 0.72 over 0.55 of MMCFSv1. Hence, MMCFSv2 improved the phase skill by 30 % and amplitude skill by about 20 %. MMCFSv2 reduced the NRMSE from 1.06 of MMCFSv1 to 0.82 (which is about 20 %) with respect to GPCP. Compared with the NMME models, MMCFSv2 has the highest skill in capturing the interannual variability of ISMR (ACC $= 0.72$). The MMCFSv2 SD is very close to observations (normalized SD $= 0.96$), and it has one of the least NRMSE values (0.82). Furthermore, the MMCFSv2's actual skill (0.72) is very close to the potential skill (0.79) and is a large improvement over MMCFSv1. MMCFSv2 has also attained the theoretical predictability limit of $\sim 0.7$. It was noticed that MMCFSv2 improves the simulated large-scale teleconnection pattern between the Niño 3.4 index and tropical SST with a higher pattern correlation of 0.70 compared with 0.63 of MMCFSv1. The spatial pattern of correlation between ERSST over the eastern Indian Ocean dipole (EIOD) box (10° S to the Equator, 90–110° E) and tropical SST anomalies has improved (pattern correlation of teleconnections from 0.31 in MMCFSv1 to 0.38 in MMCFSv2). MMCFSv2 did not reproduce the Niño 3.4-induced SST patterns over the Atlantic Ocean, whereas it was well captured by MMCFSv1. MMCFSv2 captured the eastern Indian Ocean-induced SST pattern over the tropical oceans, a pattern which was weaker in MMCFSv1.

The simultaneous correlation between the JJAS ISMR anomaly index and tropical SST anomalies showed that

both models overestimated the correlation over the western North Pacific region compared with observations. MMCFSv1 showed stronger teleconnections between ISMR and the Pacific Ocean compared with MMCFSv2 (stronger negatives over the central and eastern Pacific, and stronger positives over the western Pacific). MMCFSv2 reproduced the observed correlation patterns with a higher pattern correlation of 0.60 compared with 0.38 of MMCFSv1. Overall, MMCFSv2 captured the teleconnection between ISMR and tropical SST anomalies closer to observations than MMCFSv1.

One of the potential research areas with coupled climate models in general and MMCFSv2 is the sea surface and air temperatures biases compared to observations. The increased surface temperatures in MMCFSv2 resulted in warmer tropospheric columns in the summer hemisphere. MMCFSv2, however, simulated temperatures closer to observations in the winter hemisphere. Given that the use of MOM6 over MOM4 has enabled us to use many more parameterizations, we will address this problem in a future study. The present study's focus was to present the climatological characteristics simulated by MMCFSv2.

In conclusion, the mean state of the atmosphere has improved in MMCFSv2 (compared with MMCFSv1), both in terms of precipitation and circulation (850 hPa winds). This has resulted in improved teleconnections (Fig. 16). The pattern correlation between the spatial structure of teleconnections in Fig. 16 has improved from 0.38 in MMCFSv1 to 0.60 in MMCFSv2; hence, the interannual variability skill has improved. MMCFSv2 improves on many meteorological fields compared with MMCFSv1 in ISMR hindcasts. However, the NEMS coupling framework is the biggest improvement MMCFSv2 brings over MMCFSv1. This is central to making it easier to upgrade the individual model components as and when their respective scientific groups improve them. This is very important for an operational model.

*Code and data availability.* The current version of MMCFSv2 used for this study is available at https://doi.org/10.5281/zenodo.7905721 (Jain, 2023a) (Please check the license files for individual component model in the repository). The data used for the analysis in this paper are available at https://zenodo.org/record/7900790#.ZFU-T5FBxcA (last access: TS15; DOI: https://doi.org/10.5281/zenodo.7900790, Jain, 2023b). The complete (processed) data used to initialize and run the MMCFSv2 simulations from 1998–2022 are available at the DOIs mentioned below.

Input data from 1998 to 2000 – https://doi.org/10.5281/zenodo.7935628 (Jain, 2023c); from 2001 to 2003 – https://doi.org/10.5281/zenodo.7947318 (Jain, 2023d); from 2004 to 2006 – https://doi.org/10.5281/zenodo.7947974 (Jain, 2023e); from 2007 to 2009 – https://doi.org/10.5281/zenodo.7948155 (Jain, 2023f); from 2010 to 2012 – https://doi.org/10.5281/zenodo.7949802 (Jain, 2023g); from 2013 to 2015 – https://doi.org/10.5281/zenodo.7950855 (Jain, 2023h); from 2016–2018 – https://doi.org/10.5281/ zenodo.7949863 (Jain, 2023i); from 2019–2021 – https://doi.org/10.5281/zenodo.7950964 (Jain, 2023j); and for 2022 – https://doi.org/10.5281/zenodo.7951983 (Jain, 2023k).

Note that the original raw data belongs to NCEP (https://doi.org/10.1175/2010BAMS3001.1; Saha et al., 2010).

The documentation for CICE5 is available at https://cice-consortium-cice.readthedocs.io/en/cice6.0.0.alpha/index.html (last access: 30 December 2023). TS16

*Author contributions.* The study was conceptualized by SAR. The model deployment and simulations were carried out by DJ. The analysis was performed by DJ and RD. AS and MP helped in MMCFSv1 simulations. PAP helped in NMME data analysis. KVG carried out the heat budget analysis of the MMCFSv2 ocean model. The first draft of the manuscript was written by DJ and all authors commented on all versions. All authors approved the final manuscript.

*Competing interests.* The contact author has declared that none of the authors has any competing interests.

ther geographical representation in this paper. While Copernicus Publications makes every effort to include appropriate place names, the final responsibility lies with the authors.

*Acknowledgements.* We thank the Parthasarthee Bhatacharjee (NCEP) for helping us with the code. We thank the HPC division of IITM for the resources required to carry out the simulations and analysis of the model output. We thank the Ministry of Earth Sciences, India, for making this research possible.

*Review statement.* This paper was edited by P. N. Vinayachandran and reviewed by two anonymous referees.

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

**Remarks from the typesetter**

TS1    Please check the name. I wrote Ramu Dandi an email to confirm the first name A. Ramu and last name Dandi, but to be sure I would ask you to confirm this. Thank you.

TS2    Please contact the co-authors to update their email addresses by logging into their accounts in our system.

TS3    Please check the name with your co-author. Although I asked them to confirm your suggestion, I am also not sure if V. G. Kiran is the correct first, and Gangadharan the correct last name. Please note that at least one first name has to be written out, since initials only are not allowed according to our standards.

TS4    Please confirm this affiliation. "currently at:" is in accordance with our standards.

TS5    Please confirm citation.

TS6    Please give an explanation of why this needs to be changed. We have to ask the handling editor for approval. Thanks.

TS7    Please give an explanation of why this needs to be changed. We have to ask the handling editor for approval. Thanks.

TS8    Please confirm citations.

TS9    Please give an explanation of why this needs to be changed. We have to ask the handling editor for approval. Thanks.

TS10   Please confirm added citation.

TS11   This figure is not mentioned in the text.

TS12   Please confirm figure reference.

TS13   Please confirm.

TS14   Please confirm the citations.

TS15   Please provide date of last access.

TS16   Please confirm this section.

TS17   Please confirm addition.

TS18   Please confirm article number.

TS19   Please provide DOI/URL.

TS20   Please confirm article number.

TS21   Please confirm article number.

TS22   Please confirm article number.

TS23   Please provide date of last access.

TS24   Please provide date of last access.

TS25   Please check if this information is correct.

TS26   Please check publication year.

TS27   This reference is not mentioned in the text. There was no comment to add Iredell et al., 2014 in the text.

TS28   Please check the URL and provide date of last access.

TS29   Please provide DOI.

TS30   Please check URL and provide date of last access.