# Peer review of "Monsoon Mission Coupled Forecast System Version 2.0: Model"

_Geoscientific Model Development, 2023_

## Author Comment (AC7)

**Supplementary Figures-**

[Figure]

Figure 1a. 850hPa winds (vectors and shaded magnitude) from (a)ERA5 (b) MMCFSv1 © MMCFSv2, and biases (d) MMCFSv1 – ERA5 (e) MMCFSv2 – ERA5, difference between the model winds (f) MMCFSv2 – MMCFSv1

[Figure]

Figure 1b. 925hPa winds (vectors and shaded magnitude) from (a)ERA5 (b) MMCFSv1 © MMCFSv2, and biases (d) MMCFSv1 – ERA5 (e) MMCFSv2 – ERA5, difference between the model winds (f) MMCFSv2 – MMCFSv1

[Figure]

Figure 1c. 700hPa winds (vectors and shaded magnitude) from (a)ERA5 (b) MMCFSv1 © MMCFSv2, and biases (d) MMCFSv1 – ERA5 (e) MMCFSv2 – ERA5, difference between the model winds (f) MMCFSv2 – MMCFSv1

[Figure]

Figure 1d. 500hPa winds (vectors and shaded magnitude) from (a)ERA5 (b) MMCFSv1 © MMCFSv2, and biases (d) MMCFSv1 – ERA5 (e) MMCFSv2 – ERA5, difference between the model winds (f) MMCFSv2 – MMCFSv1

[Figure]

Figure 2a. Top panel - Standard Deviation (red), and ratio of std. Deviation to mean rainfall (color shading) from GPCP (1981-2022). Bottom panel - Variance (red), and ratio of variance to mean rainfall (color shading)

[Figure]

Figure 2b. Left panel - Variance (red), and mean rainfall (color shading) zoomed over Indian region from GPCP (1981-2022). Right panel - Variance (red), and ratio of variance to mean rainfall (color shading)

[Figure]

Figure 2c. Left panel – Standard Deviation (red), and mean rainfall (color shading) zoomed over Indian region from GPCP (1981-2022). Right panel - Standard Deviation (red), and ratio of Standard Deviation (red) to mean rainfall (color shading)

[Figure]

Figure 3. SST from (a) ERSST (b) MMCFSv1 © MMCFSv2 and biases (d) MMCFSv1 – ERSST (e) MMCFSv2 – ERSST, and model difference in SST (f) MMCFSv2 – MMCFSv1 (The contours have been adjusted to highlight 0.25 degree differences as well)

[Figure]

Figure 4a. Top panel is MLD from C-GLORS, mid panels are model bias from CGLORS and bottom panel is the difference between Mixed layer depth from MMCFSv1 and MMCFSv2.

[Figure]

Figure 4b. Heat budget analysis of mixed layer depth from MMCFSv1 and MMCFSv2 showing (a) SST differences between v2 and v1 (V2 - V1) (b) MLD difference (V2 - V1) © Qnet contribution to MLD heating.

[Figure]

Figure 5a. Scatter plot of ISMR anomaly (percentage) from GPCP (x-axis), and MMCFS (y-axis), left panel is MMCFSv1 and right panel is MMCFSv2.

| GPCP | 10% departure | | | | GPCP | 5% departure | | |
|---|---|---|---|---|---|---|---|---|
| Normal Years | V1 | False Alarm | | | Normal Years | V1 | False Alarm | |
| 19 | | 7 | | | 8 | | 4 | |
| Excess | | Hit Rate | | | Excess | | Hit Rate | |
| 3 | | 2 | | | 9 | | 10 | |
| Drought | V2 | False Alarm | | | Drought | V2 | False Alarm | |
| 3 | | 5 | | | 8 | | 2 | |
| Total Extreme Years | | Hit Rate | | | Total Extreme Years | | Hit Rate | |
| 6 | | 2 | | | 17 | | 14 | |

Table 5. Table summarizing observed normal, excess, and drought years (first column uses 10% departure from mean, and third column uses 5% departure from the mean to define extreme years). The second (10%) and the fourth column (5%) summarizes hit rates and false alarms from v1 and v2 of MMCFS.

[Figure]

Figure 6a. Global Zonal mean SST from Observation (ERSST) and models (MMCFS v1 and v2)

[Figure]

Figure 6b. 10S-10N mean SST from Observation (ERSST) and models (MMCFS v1 and v2)

[Figure]

Figure . Interannual variability of ISMR shown using anomalies in mm/day

[Figure]

Figure 8 – Monthly time series of difference in latent heat flux (lhf of 21st April initial conditions (00 and 12Z mean) minus lhf of 1st April initial conditions (00 and 12Z mean)) for 2002, 2003, and 2010 over Arabian Sea (8-16N, 54-74E).

---

## Author Response (AR1)

Dear Editor,

One of the major comments from both the referees was regarding positive SST bias over tropics in MMCFSv2. Kiran V. Gangadharan carried out the analysis of ocean surface mixed layer heat budget to come up with a preliminary explanation for the positive SST bias. This analysis is included in the revised manuscript (alongwith Figure 6). We would therefore request you to please consider adding Kiran's name to the author's list.

Thanks and Regards,
Deepeshkumar Jain et al.,
Indian Institute of Tropical Meteorology,
India

**Response to Referees' comments :**

We thank the reviewers for their useful comments, and we have incorporated these suggestions which can be addressed at this stage. One comment which was common between the two reviewers is lack of attribution of improvements to resolution, parametrization, and component of the model. It is a well-known fact that attribution of improvements/deteoration to any of the suggested component is impossible without carrying a sensitivity study, while keeping all other configurations same except that particular component. Hence, this aspect could not be addressed in this revision. However, improvements in skill of the seasonal prediction skill of ISMR could be attributed to improved mean state and its teleconnections.

**Referees Comments 1 -**

Two versions of a coupled climate model were compared regarding their prediction skill of the mean climate, the Indian summer monsoon rainfall (ISMR), and associated teleconnection. The model is Monsoon Mission CFS with version 1 and version 2. Twenty-five years of coupled seasonal hindcasts, starting in 1998, were performed from April to September. The V1 model uses 12 ensemble members, whereas the V2 model uses 10. MMCFSv1 used a horizontal resolution of T382, and that for MMCFSv2 was T574. Despite this, there are many differences between the v1 and v2 versions, including the ocean model and the coupler. The spatial and vertical structure of different fields were presented of the mean conditions and of the year-to-year co-variability. The authors state that there is an improvement in the skill of the model in predicting the seasonal mean rainfall over Indian land. This is in spite of the fact that the well known dry bias of the model did not improve from v1 to v2.

The major problem of this manuscript is it compares two different versions of the model with too many changes, including physics, the coupling method, and resolution. Previous studies have shown that changing horizontal resolution can have impacts on the model's mean conditions and teleconnection. I believe the differences between MMCFSv1 and MMCFSv2 shown in this study will fall well within the differences of changing only the resolution of the model. Thus, I suggest simulations with exactly the same resolution so that the results can be interpreted from the point of view of other components (version) and not merely from the resolution. My other specific comments are given below.

**Reply** – MMCFSv1 uses Eulerian dynamical core while MMCFSv2 uses Semi-lagrangian one. Thus, the two resolutions mentioned T382 in MMCFSv1(EL) and T574 in MMCFSv2(SL) correspond to the similar physical resolution of 1152x576 grid points in the horizontal (equivalent to ~38km resolution).

MMCFSv2 is a major upgrade (a new model even, considering the upgrades) over MMCFSv1 in terms of the framework and the component models. Hence, the manuscript can be looked at as a comparison between simulations by two different models, rather than a sensitivity study. MMCFSv2 will replace MMCFSv1 for future research works at IITM. We agree that the skill improvements as well as the limitations such as high tropospheric temperature and SST shown could have come from any of the component upgrades and this will need a thorough investigation (and a study on its own). Our focus in the present paper was to find the baseline performance of this new model in simulating ISMR and tropical climate compared to MMCFSv1. This baseline will be helpful in defining the scope of many future sensitivity studies.

Specific Comments:

**1)** Simulations used a time length of 25 years (1998-2022). It is appreciable that the authors used recent years. However, this choice does not include some critical years like 1994, 1997, and 1983 when the CFSv2 model is known to have difficulty predicting seasonal mean conditions. It is because, for example, despite an El Nino, the summer monsoon was normal due to the positive Indian Ocean Dipole.

**2)** The above comment brings to the point that a 25-year simulation is long enough to establish a particular (version) model is better than another. A short simulation misses several years of extremes and usual conditions. Other studies using CFSv2 used data sets from 1981/1982 (e.g., Ramu et al. 2016, Pillai et al. 2018). It should be possible to extend the simulations back to the available initial conditions for the robustness of the conclusions. Are the two correlation coefficients of the seasonal mean time series significantly different?

**Reply** – We would like to reply to the above two comments together here. The choice of simulation duration was made based on various factors mentioned below.

The MMCFSv2 aims to improve IMD's operational forecast by replacing the old generation model with a new one. As operational forecasts need the verification of models' performance during the recent period, we have carried out the hindcast experiment for 1998-2022. Recently, many operational centers (IMD, NCEP) have changed their climatology to include recent years. For example, NCEP-CFSv2 seasonal forecasts are now based on 1991-2020. We do agree that resolving the known problems of MMCFSv1 should be one of the major foci of future studies, however extending hindcast for such a long duration really requires lot of computational resources and the same are not available at this moment. We will address these issues in future once the required computational and storage resources are available.

Shi et al. (2014) used Ratio of Predictable Components to determine the length of hindcast simulations sufficient for studying predictability over different global regions. They showed that over the tropical regions, (including the South Asian region), a duration of 20 years is sufficient for studying hindcast predictability. Our simulation duration (25 years) satisfies this condition very well, confirming that the hindcast duration is enough to cover several of the instances covered in the comment.

MMCFSv1 is known to have difficulty in capturing critical years like 1994, 1997, and 1983. These years were characterized by EL-Nino and positive IOD. The years 2012, 2015, and 2019 from our simulations are similar El-Nino years having positive IOD. Out of these three years, MMCFSv1 could not capture 2012 and 2019, while MMCFSv2 could not capture 2019.

**3)** Most of the results were presented and discussed through eyeball comparison. For example, Fig 2 shows horizontal wind vectors at 850 hPa. The text says there is an improvement of the Somali jet in MMCFSv2 to MMCFSv1. What is the definition of the Somali Jet? At what height is it maximum, and what is the three-dimensional wind structure? It is only possible to make a conclusive statement with a definition and quantitive assessment of the phenomenon.

**Reply** – Somali Jet (most intense at 800-900hPa) is the low-level southwesterly jet over the Arabian Sea in the summer months, off the coast of Somalia. The difference in wind speeds averaged over 10-15N, 45-50E box between MMCFSv2 winds and ERA5 at 850hPa is lesser (0.09m/s) compared to that of MMCFSv1 (-0.4m/s). Similarly all the results were analyzed both qualitatively and quantitatively (with various statistics).

The difference between observed and simulated winds shown in the manuscript was used to conclude that the Findlater jet seen at 850hPa was better simulated by MMCFSv2. We have included a zoomed in picture of winds at 925, 850, 700hPa, and 500hPa (Figure a-d) with colored wind magnitude below. Both versions of MMCFS can capture this jet at 850hPa height. The difference between MMCFSv2 winds and ERA5 at 850hPa is lesser compared to that of MMCFSv1 over most of the tropical Indian ocean region.

[Figure]

Figure a. 850hPa winds (vectors and shaded magnitude) from (a)ERA5 (b) MMCFSv1 © MMCFSv2, and biases (d) MMCFSv1 – ERA5 (e) MMCFSv2 – ERA5, difference between the model winds (f) MMCFSv2 – MMCFSv1

**925hPa Winds**

[Figure]

Figure (b) Similar to Fig. (a) but for 925hPa.

**700hPa Winds**

[Figure]

Figure (c) Similar to Fig. (a) but for 700hPa.

[Figure]

Figure (d) Similar to Fig. (a) for 500hPa.

**4)** A similar argument can be given for Fig 4, where the zonal mean vertical structure of zonal wind is compared. The changes in MMCFSv2 from MMCFSv1 are both toward and away from the observations. It is better to show a pattern correlation between the total winds to demonstrate if one model is better than the other.

**Reply** – We computed the pattern correlation between the observed and simulated total zonal mean winds. Both MMCFSv1 and MMCFSv2 have a pattern correlation of 0.99. As can be inferred from this, both models have the pattern of zonal mean winds close to observations. The difference plot (Fig. 4 in manuscript) captures much more details of the simulated wind biases compared to observations.

**5)** Line 29: "The standard deviation of …  high mean precipitation." - The figure suggests that the standard deviation is high over regions with a high mean. It happens to be over the northern Bay of Bengal over south Asia. A scatter plot of mean vs variance would reveal this. In fact, over Indian land, the ratio of variance to mean will be higher than that over the north Bay of Bengal. Thus, the statement made here is not correct.

**Reply** – The ratio of variance (and standard deviation) to mean (shown in supplement) also suggests that compared to variability over oceans (including BoB), the variability over Indian land is less. This was shown in Fig. 1 of the manuscript. As suggested, we plotted variance, standard deviation, ratio of variance to mean, and ratio of standard deviation to mean. Please find these figures below (Figure e-g). Even the ratios suggest that variability over Indian land is lesser compared to oceans. Our aim in writing this (line 29) was to emphasize the difficulty faced by models in predicting the low variability of mean

ISMR over land.  We have updated Figure 1 of the manuscript to show mean, standard deviation, variance, and ratio of standard deviation and variance to mean precipitation. We have updated lines 31-32 in the Figure discussion.

[Figure]

Figure (e). Top panel - Standard Deviation (red), and ratio of std. Deviation to mean rainfall (color shading) from GPCP (1981-2022). Bottom panel - Variance (red), and ratio of variance to mean rainfall (color shading)

[Figure]

Figure (f). Left panel - Variance (red), and mean rainfall (color shading) zoomed over Indian region from GPCP (1981-2022). Right panel - Variance (red), and ratio of variance to mean rainfall (color shading)

[Figure]

Figure (g). Left panel – Standard Deviation (red), and mean rainfall (color shading) zoomed over Indian region from GPCP (1981-2022). Right panel - Standard Deviation (red), and ratio of Standard Deviation (red) to mean rainfall (color shading)

**6)** Fig 6: The difference panels (right column) show most of the bias near the equator within 2-3 degrees. Please reduce the interval of the colour scale to capture details of these biases.

**Reply** – Please find revised Fig. 6 (Figure below and Fig. 3 of the revised manuscript) in which we have adjusted the contour interval to capture more details over the equator. We would like to bring to notice that the accuracy of satellite-based SST retrieval (especially AVHRR) is 0.5 degrees.

[Figure]

Figure (h). SST from (a) ERSST (b) MMCFSv1 © MMCFSv2 and biases (d) MMCFSv1 – ERSST (e) MMCFSv2 – ERSST, and model difference in SST (f) MMCFSv2 – MMCFSv1 (The contours have been adjusted to highlight 0.5 degree differences as well)

**7)** In general, MMCFSv2 is particularly warmer than MMCFSv1 near the surface (Fig 6 and 7). A detailed reason is necessary to present to understand the changes. MMCFSv2 has several differences in modelling compared to MMCFSv1 (Table 1). Which of these components is responsible for this switch in surface temperature bias from largely cold to largely warm, especially in the northern hemisphere?

Reply - Please refer to Figure (i) and (j) below. Except over equatorial Pacific Ocean (EPO), MMCFSv1 simulates deeper mixed layer depths (MLD) compared to observations (C-GLORS). MMCFSv2 improves on this bias of MMCFSv1 with shallower MLD compared to observations, except over EPO, where the bias remains similar.

[Figure]

Figure (i). Top panel is MLD from C-GLORS, mid panels are model bias from CGLORS and bottom panel is the difference between Mixed layer depth from MMCFSv1 and MMCFSv2.

Based on the reviewers suggestion, we carried out heat budget analysis of ocean mixed layer (Figure 4) and found that the shallower MLD of MMCFSv2 is the major cause of higher SST as for a given Qnet (net energy transfer to MLD), shallower MLD will result in warmer SST. Line number 227 to 233 of the manuscript includes this discussion along with Figure 6.

[Figure]

Figure (j). Heat budget analysis of mixed layer depth from MMCFSv1 and MMCFSv2 showing (a) SST differences between v1 and v2 (b) Contribution to MLD heat from short-wave (everything increasing the heat of MLD is positive in this figure) © Long wave (d) Latent heat contribution to MLD heating (e) Long wave from ocean to MLD (f) Sensible heat contribution to MLD heating

**8)** Fig 10 suggests that overall there is an improvement (in correlation) between the model and observation in the latest version. However, there are years like 2000, 2004, and 2019 when v2's anomaly was opposite to that observed by v1's anomaly was correct. One key characteristic of a good seasonal forecast is to capture the seasonal mean extremes. A scatter plot of observation vs model would bring out if there's any difference between the two versions of the model in this respect.

**Reply** – Scatter plot Figure (k) (below) is now be added in the manuscript (Figure 11). From the scatter plot it is evident that many observed normal years were predicted as extremes in v1. Hence, we calculated the false alarm rates and the hit rates for both the models. We used two criteria for defining normal years, viz 10% and 5% departure from the climatological mean. Table (k) below summarizes the false alarms and hit rates. As seen from the table, MMCFSv1 has a higher false alarm rate and a lower hit rate than MMCFSv2.

[Figure]

Figure (k). Scatter plot of ISMR anomaly (percentage) from GPCP (x-axis), and MMCFS (y-axis), left panel is MMCFSv1 and right panel is MMCFSv2.

| GPCP | 10% departure | | | GPCP | 5% departure | |
|---|---|---|---|---|---|---|
| Normal Years | V1 | False Alarm | | Normal Years | V1 | False Alarm |
| 19 | | 7 | | 8 | | 4 |
| Excess | | Hit Rate | | Excess | | Hit Rate |
| 3 | | 2 | | 9 | | 10 |
| Drought | V2 | False Alarm | | Drought | V2 | False Alarm |
| 3 | | 5 | | 8 | | 2 |
| Total Extreme Years | | Hit Rate | | Total Extreme Years | | Hit Rate |
| 6 | | 2 | | 17 | | 14 |

Table (k) Table summarizing observed normal, excess, and drought years (first column uses 10% departure from mean, and third column uses 5% departure from the mean to define extreme years). The second (10%) and the fourth column (5%) summarizes hit rates and false alarms from v1 and v2 of MMCFS.

**9)** In Table 3, how is the correlation over the western parts of the equatorial Indian Ocean?

**Reply** – In both these models the western parts of eq. Indian Ocean are positively correlated. MMCFSv1 skill (SST) over west IOD box (10S-10N, 50-70E) is 0.44. MMCFSv2 skill (SST) over west IOD box (10S-10N, 50-70E) is 0.40.

**10)** It is unclear what is the reason behind improved skill (interannual correlation of seasonal mean). Does it come because of a better simulation of climate patterns or its teleconnection to the monsoon?

**Reply** – The mean state of the atmosphere has improved, both in terms of precipitation and circulation (850hPa winds). This has resulted in improved teleconnections (Figure 18 of revised manuscript). The pattern correlation between spatial structure of teleconnections (in Figure 18 of the revised manuscript) has improved from 0.38 in MMCFSv1 to 0.60 in MMCFSv2. Hence the interannual variability has improved.  This discussion has been added to the manuscript at the end of the discussion section (Lines 535-541).

**Referees Comments 2 :**

This study introduced the Monsoon Mission Coupled Forecast System Version 2.0 and compared its hindcast results with the previous version for the recent 25 years. The MMCFS v2 simulates better tropical wind and rainfall compared to the v1 model, while the temperature fields became worse. MMCFSv2 captures significant features of the Indian monsoon, including the intensity and location of the maximum precipitation centers and the large-scale monsoon circulation. The 25-y hindcast results from the v2 model are compared with that from v1, and found that the v2 model improves the simulation skill in rainfall pattern and amplitude.

  This manuscript is titled with a model description, but there is no detailed information about the key model configurations. There is no information regarding what has been changed compared to the original component model, or the v2 model is just an integration of existing component models. The manuscript described the basic performance of the v2 model in simulating the mean states and interannual variation, while the reason for these improvements is not well present.

**Reply** –Details are provided in Tabel1 (revised) and the corresponding description in the manuscript.

**Major comments:**

**1)** Based on the description of the MMCFSv2 model, it coupled the MOM6, GFS-SL, and CICE5 together. Is there any model tuning before the hindcast simulation?

**Reply** – There was no tuning done before carrying out the simulations. These are the very first hindcasts which will guide us in tuning the model in future simulations. This dataset will be the baseline for future sensitivity studies with MMCFSv2. The first of these will be correcting the SST and temperature bias.

**2)** The evaluation of the mean states shows bias in SST, circulation, and precipitation. The linkage among those biases should be discussed in detail, especially for the v2 model.

**Reply** – The biases in SST comes from shallower MLD simulated by MMCFSv2 (Figure (i) and (j) of this document). The improved circulation of MMCFSv2 is most likely the result of improved convective centers from MMCFSv1 (Figure 9 of original manuscript, Fig. 2 of revised manuscript). We would like to say that establishing the linkages (cause and effect) in a highly non-linear coupled model is not trivial and is a study on its own. Our focus with the present manuscript is to document the performance of MMCFSv2 and to explore the opportunities for future research.

**3)** There are large differences between the v1 and v2 models. What is the major cause of those changes?

**Reply** – The biggest change, which we believe, has contributed to the most difference is MOM6 ocean model. MOM6 is running at higher resolution than MOM4.  Improvements brought by MOM6 over MOM4 include using C-grid stencil over B-grid stencil. C-grid stencil is preferred for simulations involving an active mesoscale eddy field. MOM6 uses scale-aware parameterizations for mesoscale eddy-permitting regimes. As we see from the results, MOM6 produces significantly different SST patterns. The difference in SST pattern is the result of shallower MLD. The better winds are the result of better

convective centers of precipitation. The better ISMR skill in the model is the result of better teleconnections (Fig. 18 of the revised manuscript). Though, it is impossible to attribute the improvements to any (particular) component of the model as all the components are coupled to each other and as a result, biases in one component can influence the other component.

4) The evaluation of the mean states of the MMCFS v2 show that it has a larger bias in the SST and surface temperature compared to the MMCFS v1 model. However, the v2 model shows better performance in simulating the 850 hPa and 200 hPa circulations and precipitation. Why and how can the v2 improve precipitation with degradation in SST and surface temperature?

**Reply** – If we only consider the SST magnitudes, then there is a degradation in MMCFSv2 simulations. However, the SST gradients (both zonal and meridional) show similar structures between v1 and v2. Please see the mean SST gradients below (Figures (l-m). As much as the overall magnitudes, we also believe gradients and deep moist convection plays a significant role in establishing global circulation patterns (Lindzen and Nigam 1987; Back and Bretherton 2009; Wallace 1989; Chelton et al. 2004). We also found that the convection centers are better simulated in MMCFSv2. This has resulted in better

[Figure]

Figure (l). Global Zonal mean SST from Observation (ERSST) and models (MMCFS v1 and v2)

[Figure]

Figure (m). 10S-10N mean SST from Observation (ERSST) and models (MMCFS v1 and v2)

850hPa winds.

5) A skillful seasonal prediction relies on reliable data assimilation for the initial conditions. The ICs are all obtained from NCEP CFSR. Does the CFSR have the same component models and the same resolutions as MMCFS v2? What is the difference between the CFSR and MMCFS v2 in terms of model configurations? Do these two models share similar model performance and bias? If the two models have different mean states, why can the ICs from CFSR be used in the v2 model? What is the impacts of initial shock and model drift on the hindcast results?

**Reply** - CFSR uses the same model configuration as MMCFSv1 (viz. MOM4, GFS-EL, SIS). MMCFSv2 has an upgraded setup (MOM6, GFS-SL, and CICE5) compared to CFSR. We re-gridded the ocean initial conditions from CFSR to MOM6 grids. We also re-gridded atmospheric model initial conditions for using them in GFS-SL.

The effects of initial shock in MMCFSv1 were studied by Shukla et al., (2018). They used Latent heat flux over Arabian sea as one of the major variable for their analysis. Carrying out a similar analysis is not possible with current MMCFSv2 setup. However, to address reviewers query, we have compared the effect of initial shock with 2 ensemble mean. Please note that we will not include this in the manuscript as this is not the complete analysis.

We took three years, viz 2002 (deficit), 2003 (normal), and 2010 (excess) ISMR years. We took the mean of 1st April 00 and 12Z initial condition simulations and computed the difference from 00 and 12Z of 21st April initial conditions for Latent heat flux over Arabian Sea (8-16N, 54-74E). To our surprise, the difference (Figure below) shows a larger initial shock in LHF in MMCFSv1 compared to MMCFSv2.

[Figure]

Figure (n) – Monthly time series of difference in latent heat flux (lhf of 21st April initial conditions (00 and 12Z mean) minus lhf of 1st April initial conditions (00 and 12Z mean)) for 2002, 2003, and 2010 over Arabian Sea (8-16N, 54-74E).

**6)** In Fig. 11, please also show the simulated ISMR anomaly for the v1 and v2 models. Based on this limited time period hindcast results, it is hard to say which one is better. In Fig. 11 and L285-286, by comparing the blue and purple bars, MMCFSv2 has the correct sign for 19 years. And MMCFS v1 predicted the correct sign for 18 years. The climate impacts for the extreme years (e.g., anomaly exceeding 10%) are more significant. A better prediction is more valuable for extreme years than normal years. For those extreme dry (e.g., 2002, 2004, 2009, 2015) and wet (2019, 2020) years, the simulation from the v1 model looks better than the v2 model, as shown in Fig. 11.

**Reply** - Figure below shows the interannual variability of observed and simulated ISMR anomalies. Yes, for years 2000, 2004, and 2019, the v2 anomaly was opposite to that observed and v1 was correct. Conversely, for 2008, 2011, 2017, and 2018, v2 got the sign correctly and v1 was the opposite. The attached scatter plot in Figure (k) (percentage departure) highlights the years mentioned here as well as the other extreme years such as 2007 (v2 is better than v1), 2010 (v1 and v2 gets similar result), 2002, and 2009. Figure (k) shows a lot of normal years which were wrongly predicted as extreme years in MMCFSv1. Hence, we calculated the false alarm rates and the hit rates for these years. We used two criteria for defining normal years, viz 10% and 5% departure from the climatological mean. Table (k) summarizes the false alarms and hit rates. As seen in the table (Table 3 of revised manuscript), MMCFSv1 has a higher false alarm rate and a lower hit rate than MMCFSv2.

[Figure]

Figure (o). Interannual variability of ISMR shown using anomalies in mm/day

**7)** The authors claim that the v2 model has better skills in simulating the ISMR. However, why can the v2 model do better in the hindcast experiment? This physical explanation is missing in the current manuscript.

**Reply** - The mean state of the atmosphere has improved, both in terms of precipitation and circulation (850hPa winds). This has resulted in improved teleconnections (Figure 18 of revised manuscript). The pattern correlation between spatial structure of teleconnections in Figure 18 (revised manuscript) has improved from 0.38 in MMCFSv1 to 0.60 in MMCFSv2. Hence the interannual variability skill has improved. This discussion has been added to the manuscript at the end of the discussion section.

Minor comments:

**8)** It is better to discuss the temperature bias first and then consider its impact on precipitation and circulation. In section 4.1.1, why is an improvement in circulation? Is it due to the changes in model physics or model resolution?

**Reply** – We thank the referee for this. This will improve the presentation of results significantly. We have rearranged the discussion so that temperature biases are discussed before the discussion on circulation.

**9)** L114-115. The initialization of the two versions of models needs to be further documented. The MMCFSv1 and v2 have different resolutions. Does the CFSR system provide all the initial conditions for the v1 and v2 resolution? Is there any data assimilation in preparing the initial condition by the authors?

**Reply** – Same CFSR initial conditions were used for both the models. We have not done any data assimilation from our side before carrying out these simulations other than regridding.

**10)** L125 What are the initial dates for the 12-member prediction in MMCFSv1? It would be better to introduce it here than refer to a paper.

**Reply** – The initial dates for these simulations were similar between v1 and v2 (mentioned in the manuscript), except that v1 had two additional ensembles starting from 00z and 12z of 26th April.

**11)** In Fig. 6, where is the 0.5K contour line?

**Reply** – We have updated the Figure with contour lines. Please see Figure 3 of the manuscript.

**12)** Fig. 6-8 show the MMCFS v2 model has a large bias compared to the previous one. Is this due to the energy bias in the AGCM? Or the problem in air-sea coupling? It is better to reduce this apparent mean bias before the prediction.

**Reply** – All the coupled climate models have biases and it is impossible to carry out simulations only after the biases have been reduced/removed. CMIP6 models also suffer from similar warm SST biases (Zhang et al. 2023) as MMCFSv2. The warm SST bias is one of the first bias which will be addressed in future, for MMCFSv2. The warmer SST in MMCFSv2 can be explained in terms of the shallower MLD simulated by the model for a given Qnet (Figure (i) and (j)). This discussion is included in the revised manuscript (Figure 6)

**13)** Table 3, please add the significant test for these numbers. What are the definitions of these modes?

**Reply** –We have highlighted values with 95% significance. We have also included the definitions of these modes.

**14)** L396, 'Fig. 14(c)' should be Fig. 14(b).

**Reply** – Corrected in the revised manuscript.

**15)** 'MMCFSv2 captures these teleconnection patterns over the tropical Oceans and the eastern Indian Ocean (Fig. 14 (c)).'. This is not true.

Reply – Corrected in the revised manuscript.

**16)** L403-414, Fig. 15 did not describe the impact of the IOD. Please do not use IOD in the context.

Reply – Though Fig. 15 only shows the effects of EIOD and is discussed here, Table 3 summarizes the effects of IOD on ISMR and is also discussed here.

**17)** L444-445, based on Fig. 6-8. This is not true for SST and surface temperature.

Reply – L444-445 has been updated to reflect the same.

**References -**

1) Shi, W., et al. "Impact of hindcast length on estimates of seasonal climate predictability." Geophysical research letters 42.5 (2015): 1554-1559.

2) Sridevi, Ch, et al. "Rainfall forecasting skill of GFS model at T1534 and T574 resolution over India during the monsoon season." Meteorology and Atmospheric Physics 132 (2020): 35-52.

3)Shukla, R.P., Huang, B., Marx, L. et al. Predictability and prediction of Indian summer monsoon by CFSv2: implication of the initial shock effect. Clim Dyn 50, 159–178 (2018). https://doi.org/10.1007/s00382-017-3594-0

4) Zhang, Qibei, et al. "Understanding models' global sea surface temperature bias in mean state: from CMIP5 to CMIP6." Geophysical Research Letters 50.4 (2023): e2022GL100888.